# SHADOWKV: KV Cache in Shadows for High-Throughput Long-Context LLM Inference

Hanshi Sun [1 2]   Li-Wen Chang [1]   Wenlei Bao [1]   Size Zheng [1]   Ningxin Zheng [1]   Xin Liu [1]   Harry Dong [2]
Yuejie Chi [2]   Beidi Chen [2]

## Abstract

With the widespread deployment of long-context large language models (LLMs), there has been a growing demand for efficient support of high-throughput inference. However, as the key-value (KV) cache expands with the sequence length, the increasing memory footprint and the need to access it for decoding both result in low throughput when serving long-context LLMs. While various dynamic sparse attention methods have been proposed to accelerate inference while maintaining generation quality, they either fail to sufficiently reduce GPU memory usage or introduce significant decoding latency by offloading the KV cache to the CPU. We present SHADOWKV, a high-throughput long-context LLM inference system that stores the low-rank key cache and offloads the value cache to reduce the memory footprint for larger batch sizes and longer sequences. To minimize decoding latency, SHADOWKV employs an accurate KV selection strategy that reconstructs minimal sparse KV pairs on-the-fly. By evaluating SHADOWKV on benchmarks like RULER, LongBench, and models such as Llama-3.1-8B and GLM-4-9B-1M, we demonstrate that it achieves up to 6× larger batch sizes and 3.04× higher throughput on an A100 GPU without sacrificing accuracy, even surpassing the performance achievable with infinite batch size under the assumption of infinite GPU memory.

## 1. Introduction

Large language models (LLMs) have increasingly demonstrated their ability to scale and handle long contexts

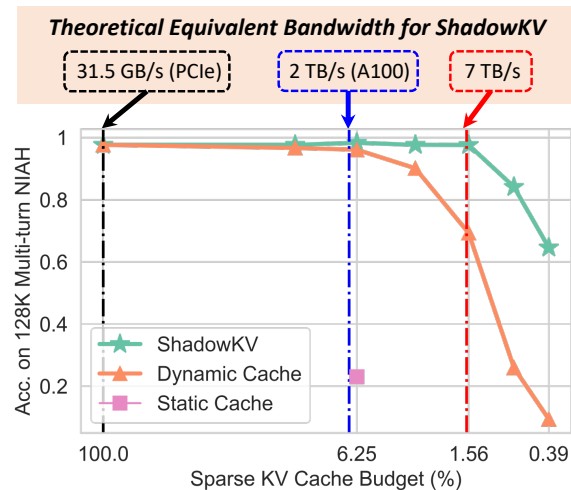

Figure 1. SHADOWKV effectively utilizes a limited KV budget to achieve high accuracy, theoretically reaching over 7 TB/s equivalent bandwidth on an A100 GPU.

(Achiam et al., 2023; Team et al., 2023; Microsoft, 2024; Liu et al., 2024a), enabling them to tackle complex tasks like multi-document question answering and information retrieval from extensive contexts of up to 1M tokens (Achiam et al., 2023; Wang et al., 2024b). However, efficiently serving these long-context LLMs presents challenges related to the key-value (KV) cache (Ge et al., 2023; Liu et al., 2024b), which stores previous key-value activations to avoid re-computation. As KV cache scales with sequence length, its growing memory footprint and the need to access it for each token generation lead to low throughput during long-context LLM inference. To address these, KV eviction or sparse attention methods have been widely explored.

However, existing methods face three primary limitations: accuracy degradation, inadequate memory reduction, and significant decoding latency overhead. KV cache eviction strategies (Zhang et al., 2024d;c) aim to reduce the memory footprint by discarding KV pairs based on specific policies, but they often result in information loss and accuracy degradation in tasks such as multi-turn conversations (Yang

---

[1]ByteDance Seed [2]Carnegie Mellon University. Correspondence to: Hanshi Sun <hanshi.s@bytedance.com>, Beidi Chen <beidic@andrew.cmu.edu>.

*Proceedings of the 42nd International Conference on Machine Learning*, Vancouver, Canada. PMLR 267, 2025. Copyright 2025 by the author(s).

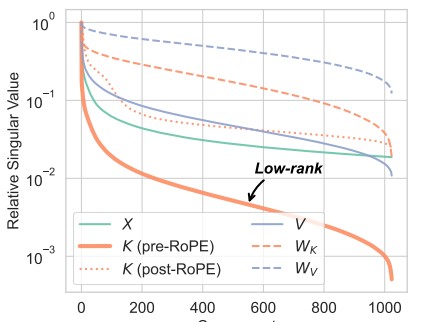 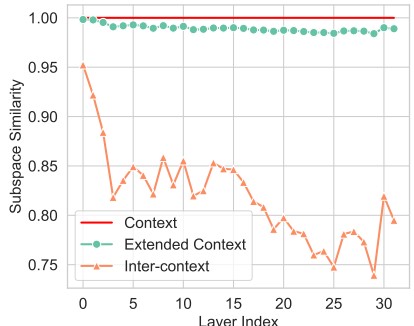 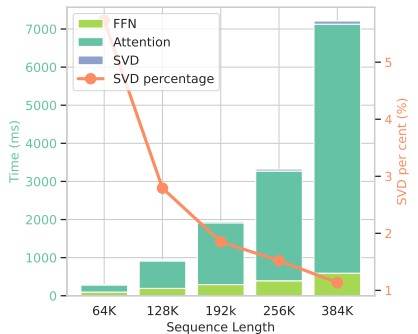

*Figure 2.* **Left:** For a sample from PG-19 (Rae et al., 2019) fed into Llama-3.1-8B, the pre-RoPE keys are the most low-rank, as indicated by the sharpest decay in singular values. **Middle:** Average similarities, defined in Section 3.1, between rank-256 truncated SVD projections of pre-RoPE keys from PG-19 sequences using Llama-3.1-8B. Similarity is measured between a length 16K "Context" and either a 16K+2K continuation on "Context" ("Extended context") or a new length 16K sequence ("Inter-context"). Pre-RoPE keys share low-rank subspaces within sequences but differ across sequences. **Right:** The relative overhead of singular value decomposition (SVD) decreases as sequence length scales for the pre-filling stage.

et al., 2024b; Tang et al., 2024a). Dynamic sparse attention methods (Tang et al., 2024b) preserve all KV pairs on the GPU and accelerate inference by computing attention with selected KV pairs. However, this line of work does not mitigate the memory footprint, thereby limiting the batch size and preventing accommodation of extremely long contexts (e.g., 1M tokens). A naive solution based on sparse attention involves offloading the KV cache to the CPU to reduce memory usage (Lee et al., 2024a; He & Zhai, 2024). Nonetheless, as shown in Figure 4, this approach incurs significant overhead due to the latency of fetching the selected sparse KV pairs from the CPU during decoding.

Consequently, an ideal system for long-context LLM inference with sparse attention should: (i) reduce GPU memory usage, (ii) minimize inference latency, and (iii) maintain accuracy within limited sparse KV cache budgets. Fortunately, we can potentially overcome these challenges by leveraging our discovery that pre-Rotary Position Embedding (Su et al., 2024) (RoPE) keys are exceptionally low-rank compared to the layer inputs, post-RoPE keys, values, key weight matrix, and value weight matrix, as indicated in Figure 2. Unlike prior works that apply low-rank approximations to weights (Chang et al., 2024; Lee et al., 2024b), we directly compress pre-RoPE key cache, achieving higher accuracy. Our analysis reveals that pre-RoPE keys lack significant similarities in low-rank subspaces across different sequences, while a sequence and its continuation tend to strongly share low-rank subspaces, enabling high compression rates within each sequence. Motivated by these findings, we developed two key insights that pave the way for the design of an applicable system, detailed in Section 3.

*Low-rank Keys and Offloaded Values for Storage*: In long-context LLM inference, the quadratic scaling of attention computation with sequence length makes the linear cost

of low-rank decomposition during pre-filling negligible, as illustrated in Figure 2[1]. To reduce memory footprint, we retain the low-rank pre-RoPE key cache on the GPU and offload the value cache to the CPU since the value cache does not exhibit low-rank properties, minimizing memory footprint without sacrificing accuracy. During decoding with sparse attention, we employ CUDA multi-streams to overlap the recovery of the selected key cache with the fetching of the corresponding value cache. This approach conceals key cache reconstruction and reduces data fetching overhead by 2× compared to the naive offloading strategy, thereby decreasing the decoding latency of sparse attention.

*Accurate KV Selection for Fast Decoding*: To further reduce decoding latency in sparse attention, we propose an accurate KV selection method that maintains accuracy with minimal number of selected tokens (i.e., the K of TopK), which we refer to as sparse budgets (1.56%). Our analysis reveals that most post-RoPE keys exhibit high cosine similarity with adjacent tokens, enabling chunk-level approximations for selecting important tokens. A minimal number of outlier chunks (0.3%), which are more challenging to approximate (Figure 5), are stored as static cache on the GPU to preserve accuracy. As shown in Figure 1, our method outperforms the naive sparse attention approach (Tang et al., 2024b) and achieves higher sparsity, accelerating decoding.

Building on these insights, we present SHADOWKV in Section 4, depicted in Figure 3, a high-throughput system for long-context LLM inference. Specifically, during pre-filling, we offload the value cache to the CPU, retaining only the low-rank pre-RoPE keys, along with compressed landmarks of the key cache and detected outliers for larger batch sizes.

---

[1]In practical scenarios, the key cache can be offloaded to the CPU to perform SVD asynchronously or precomputed and stored as part of the prefix cache (Juravsky et al., 2024).

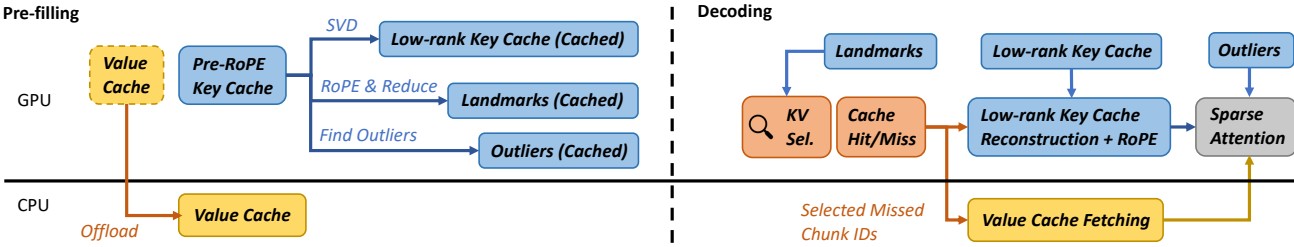

*Figure 3.* SHADOWKV enhances long-context LLM inference throughput by offloading the value cache to the CPU while maintaining a low-rank key cache, landmarks, and outliers on the GPU. During decoding, it employs landmarks for efficient sparse attention, reducing computation and data movement.

During decoding, landmarks are used to select chunk indices for key cache recovery and value cache fetching. We perform accurate sparse attention computation with selected KV pairs and static outliers to achieve high throughput.

Empirically, we conduct extensive experiments and ablation studies to demonstrate the effectiveness and efficiency of SHADOWKV. In Section 5.1, we evaluate across various long-context LLMs, such as Llama-3-8B-1M (Gradient., 2024), Llama-3.1-8B (Meta AI, 2024), GLM-4-9B-1M (GLM et al., 2024), Yi-9B-200K (AI et al., 2024), Phi-3-Mini-128K (Abdin et al., 2024) and Qwen2-7B-128K (Yang et al., 2024a) using benchmarks including RULER (Hsieh et al., 2024), LongBench (Bai et al., 2023), and Needle In A Haystack (Kamradt, 2023) with contexts up to 1M.

In Section 5.2, we demonstrate that SHADOWKV can support 6× larger batch sizes and boost throughput by up to 3.04× compared to small batches on an A100 using Llama-3.1-8B. We also present results across different models and context lengths, increasing throughput up to 2.97× for Llama-3-8B-1M, 2.56× for GLM-4-9B-1M, and 2.66× for Yi-9B-200K, even surpassing infinite batch size under the assumption of infinite GPU memory. The code is available at `https://github.com/ByteDance-Seed/ShadowKV`.

## 2. Related Works

**Token Eviction.** To reduce memory footprint, eviction-based strategies keep a fixed size of KV cache to store the critical token KV pairs and discard unnecessary tokens. StreamingLLM (Xiao et al., 2023b) addresses the limitations of window attention by retaining attention sinks and recent KV pairs. $H_2O$ (Zhang et al., 2024d) introduces a low-cost eviction policy, updating the KV cache based on cumulative attention scores. LESS (Dong et al., 2024b) accumulates evicted token information by a constant-sized low-rank cache, which allows partial access to evicted information, along with tokens maintained by a sparse policy. SnapKV (Li et al., 2024) uses the local window of prompts to select important tokens for future generations. However,

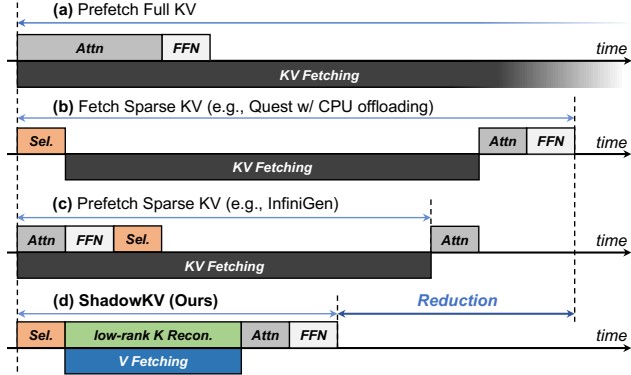

*Figure 4.* SHADOWKV outperforms prior works (Tang et al., 2024b; Lee et al., 2024b) and scales better to larger KV.

they suffer from performance degradation and information loss since the evicted tokens will never be recovered.

**Dynamic Sparse Attention.** This line of work retains all KV cache but performs dynamic sparse attention within selected KV pairs to reduce inference latency. SparQ (Ribar et al., 2023) uses the norm of the query to decide an important subset of the key cache's channels to calculate a metric to select relevant tokens. Quest (Tang et al., 2024b) segments tokens into pages and selects pages by approximating the highest attention within each page. Loki (Singhania et al., 2024) performs principal component analysis on key caches using a calibration dataset, selecting tokens based on attention scores computed in low-dimensional space. Tri-Force (Sun et al., 2024a) combines sparse attention with speculative decoding (Leviathan et al., 2023) for lossless acceleration. InfiniGen (Lee et al., 2024a) offloads the entire KV cache to the CPU and prefetches essential entries using predefined projections via SVD for KV selection. In contrast, SHADOWKV employs an online, prompt-dependent SVD for key cache compression, reducing data fetching. As shown in Figure 4, SHADOWKV scales better, as overlapping KV fetching and computation becomes challenging with larger KV caches. Additionally, InfiniGen suffers performance drops due to inaccurate prefetching KV selection.

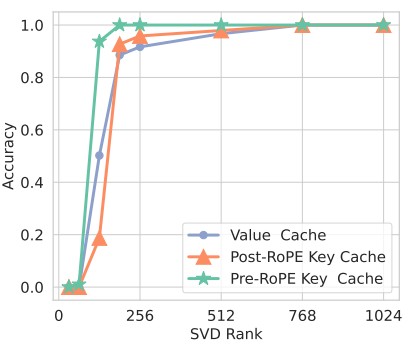 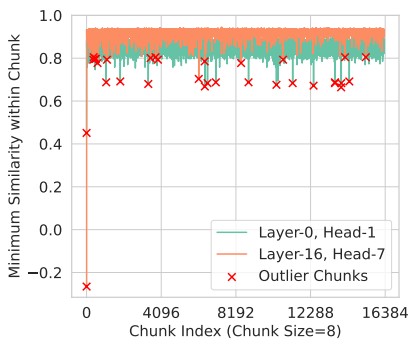 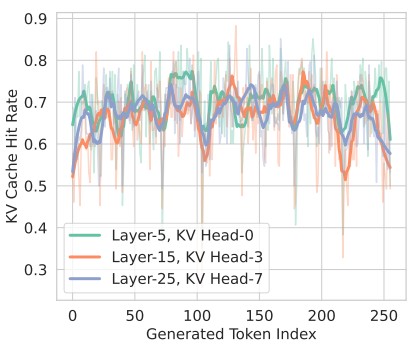

*Figure 5.* **Left:** Accuracy on the needle retrieval across various ranks shows that the pre-RoPE key cache can be compressed by over 6 times without a drop in accuracy. **Middle:** The number of notable outlier chunks is small, taking only 0.2-0.3%. **Right:** The KV cache has a high hit rate, reducing computations and data movements by over 60% for each decoding step.

**Quantization.** Several methods have been introduced to optimize KV cache quantization (Hooper et al., 2024; Yue et al., 2024; Xiao et al., 2023a), reducing memory consumption while retaining accuracy. KIVI (Liu et al., 2024c) applies different quantization strategies for keys and values, quantizing the keys per-channel and the values per-token to 2-bit. Palu (Chang et al., 2024) decomposes KV weight matrices offline, caching low-rank KV projections to achieve a higher compression rate. Quantization methods reduce the KV cache bit width, which is orthogonal to our approach.

## 3. Observations

We present two key insights of long-context LLMs that inspire SHADOWKV's design, as follows.

### 3.1. Low-Rank Keys and Offloaded Values for Storage

To reduce memory footprint, the low-rank nature of the KV cache has been explored by recent studies (DeepSeek-AI, 2024; Xu et al., 2024; Chang et al., 2024; Saxena et al., 2024; Yu et al., 2024; Lin et al., 2024). However, these methods focus on data-independent decomposition, either requiring training or achieving limited compression rates.

**Observation.** In our study, we visualize the relative singular value distributions of the model weights $W_k$, $W_v$, the input $X$, the pre-/post-RoPE key cache, and the value cache of Llama-3.1-8B by conducting SVD, as shown in Figure 2. In Figure 5, we further analyze the accuracy impact. We observe that pre-RoPE keys exhibit the lowest rank, allowing for $6\times$ compression without performance degradation.

We also identify striking dynamic and static behaviors in low-rank keys between and within sequences, inspired by a related investigation in FFN layers (Dong et al., 2024a). Analogous to cosine similarity, we define $\mathcal{D}(\boldsymbol{H}_1, \boldsymbol{H}_2) = \langle \boldsymbol{H}_1, \boldsymbol{H}_2 \rangle / r$ to be the similarity metric between low-

rank subspaces of two rank-$r$ projection matrices, $\boldsymbol{H}_1$ and $\boldsymbol{H}_2$, where $\langle \cdot, \cdot \rangle$ is the Frobenius inner product[2]. In our case with truncated SVDs of pre-RoPE keys, let $\boldsymbol{K}_1, \boldsymbol{K}_2 \in \mathbb{R}^{n \times d}$ have rank-$r$ truncated SVDs, $\boldsymbol{\Phi}_1 \boldsymbol{\Sigma}_1 \boldsymbol{\Psi}_1^\top$ and $\boldsymbol{\Phi}_2 \boldsymbol{\Sigma}_2 \boldsymbol{\Psi}_2^\top$, respectively, where $\boldsymbol{\Phi}_1 \in \mathbb{R}^{n \times r}, \boldsymbol{\Sigma}_1 \in \mathbb{R}^{r \times r}, \boldsymbol{\Psi}_1 \in \mathbb{R}^{d \times r}$, and similarly for $\boldsymbol{\Phi}_2$, $\boldsymbol{\Sigma}_2$, and $\boldsymbol{\Psi}_2$. Then, $\mathcal{D}(\boldsymbol{\Psi}_1 \boldsymbol{\Psi}_1^\top, \boldsymbol{\Psi}_2 \boldsymbol{\Psi}_2^\top)$ can measure the similarity between the low-rank subspaces of the two right singular matrices. Depicted in Figure 2, pre-RoPE keys between sequences do not strongly share similar low-rank subspaces, but extensions of the same sequence do. Thus, applying low-rank approximations to weights alone degrades performance.

**Insights.** Our observation of the low-rank nature in the pre-RoPE keys indicates that storing the low-rank projections is sufficient for each sequence. By keeping the low-rank key cache on the GPU and offloading the value cache to the CPU since it is not low-rank, we can largely reduce the memory footprint. During decoding, selected KV pairs can be reconstructed on-the-fly for computation.

### 3.2. Accurate KV Selection for Fast Decoding

To further reduce the latency overhead in sparse attention, including fetching the selected value cache from the CPU and reconstructing the corresponding key cache, an accurate KV selection method is needed to minimize the sparse KV cache budget while maintaining the accuracy.

**Observation.** We found most post-RoPE key cache exhibits spatial locality, with high cosine similarity to adjacent

---

[2]Since $\boldsymbol{H}_1$ and $\boldsymbol{H}_2$ are projection matrices, their squared Frobenius norms are the sum of their singular values which consist of $r$ 1's and $d-r$ 0's, i.e., $\|\boldsymbol{H}_1\|_F^2 = r$. Thus, by Cauchy-Schwarz, $|\mathcal{D}(\boldsymbol{H}_1, \boldsymbol{H}_2)| \le 1$. Additionally, $\mathcal{D}(\boldsymbol{H}_1, \boldsymbol{H}_2) \ge 0$ by the cyclic property of trace and positive semidefiniteness of projection matrices. Together, this shows $\mathcal{D}(\boldsymbol{H}_1, \boldsymbol{H}_2) \in [0, 1]$, maximized or minimized when the projection matrices project onto identical or orthogonal subspaces, respectively.

**Algorithm 1** SHADOWKV Pre-filling

**Input:** $K, K^{\text{RoPE}}, V \in \mathbb{R}^{b \times h_{kv} \times s \times d}$, SVD rank $r$, chunk size $c$, number of outlier chunks $o$

▷ *Store low-rank projection of pre-RoPE key cache*
$A \in \mathbb{R}^{b \times s \times r}, B \in \mathbb{R}^{b \times h_{kv} \times r \times d} \leftarrow \text{SVD}(K)$

▷ *Segment post-RoPE key cache into chunks and compute the mean of each chunk*
$C \in \mathbb{R}^{b \times h_{kv} \times s/c \times d} \leftarrow \text{Reduce}(K^{\text{RoPE}})$

▷ *Compute cosine similarity within each chunk*
$S \in \mathbb{R}^{b \times h_{kv} \times s/c \times c} \leftarrow \text{CosineSimilarity}(C, K^{\text{RoPE}})$

▷ *Find lowest cosine similarity as outliers*
$I \in \mathbb{R}^{b \times h_{kv} \times o} \leftarrow \text{ArgTopK}(-\text{Min}(S, \text{dim} = -1), o)$
$K^{\text{outlier}}, V^{\text{outlier}} \leftarrow \text{Gather}(K^{\text{RoPE}}, V, I)$

▷ *Offload the rest of values to the CPU and store the non-outlier chunks' mean as landmarks*
$V^{\text{CPU}} \leftarrow V \setminus V^{\text{outlier}}; L \leftarrow C \setminus \text{Gather}(C, I)$

---

**Algorithm 2** SHADOWKV Decoding

**Input:** $A, B, L, V^{\text{CPU}}, Q \in \mathbb{R}^{b \times h_q \times s_q \times d}, K^{\text{outlier}}$, $V^{\text{outlier}}, K, V \in \mathbb{R}^{b \times h_{kv} \times s_q \times d}$, number of chunks $n_c$, number of selected chunk budget $k$

▷ *Compute chunk attention score*
$P \in \mathbb{R}^{b \times h_q \times s_q \times n_c} \leftarrow \text{MatMul}(Q, L^{\top})$
$S \in \mathbb{R}^{b \times h_q \times s_q \times n_c} \leftarrow \text{Softmax}(P/\sqrt{d})$
$S_1 \in \mathbb{R}^{b \times h_q \times n_c} \leftarrow \text{sum}(S, \text{dim} = -2)$
$S_2 \in \mathbb{R}^{b \times h_{kv} \times n_c} \leftarrow \text{max}_{\text{kv\_group}}(S_1)$

▷ *Select top-k chunks for each KV head*
$I \in \mathbb{R}^{b \times h_{kv} \times k} \leftarrow \text{ArgTopK}(S_2, k)$

▷ *Gather value cache from the CPU*
$V^{\text{sparse}} \leftarrow \text{Gather}(V^{\text{CPU}}, I); V \leftarrow [V^{\text{outlier}}; V^{\text{sparse}}; V]$

▷ *Reconstruct key cache from low-rank projection*
$K^{\text{sparse}} \leftarrow \text{MatMul}(\text{Gather}(A, I), B)$
$K \leftarrow [K^{\text{outlier}}; \text{RoPE}(K^{\text{sparse}}); K]$

---

tokens, except for a few outliers. To quantify this, we conducted inference experiments on 128K contexts. We divided the post-RoPE keys into chunks of eight tokens and visualized the minimum cosine similarity between the chunk's mean and its key cache. The results indicate that, apart from a few outliers, there is generally high cosine similarity, suggesting the mean values can serve as landmarks to approximate attention well within normal chunks.

**Analysis.** This finding suggests that for the majority of chunks, we can maintain the mean value as compressed landmarks to select minimal important KV pairs (1.56%) accurately during decoding. Outlier chunks, which may contain dense or critical information and are difficult to approximate, are retained to ensure accuracy. Given their relatively small number (0.2–0.3%), storing them on the GPU is feasible without affecting memory capacity. Furthermore, as shown in Figure 5, considering the temporal locality of the KV cache—meaning that the KV pairs selected by the queries of two adjacent decoding steps have a high repetition rate, a cache policy (Zhang et al., 2024a) can be leveraged to further reduce the latency overhead by 60% during decoding with optimized CUDA kernels.

## 4. SHADOWKV

In this section, we introduce SHADOWKV, a novel high-throughput long-context LLM inference system. We first elaborate our algorithm in Section 4.1, covering both the pre-filling and decoding phases. Subsequently, in Section 4.2, we discuss the concept of theoretical equivalent bandwidth to illustrate the benefits of our approach.

### 4.1. Algorithm

The algorithm of SHADOWKV is divided into two main phases: pre-filling and decoding. The pre-filling phase in-

volves low-rank decomposition of the post-RoPE key cache, offloading the value cache, and constructing landmarks to facilitate subsequent high-throughput decoding. The decoding phase includes accurate KV selection and efficient sparse KV cache reconstruction.

**Pre-filling.** During the pre-filling phase, we optimize GPU memory usage by performing low-rank compression on the key cache of each layer and offloading values to the CPU. Specifically, as demonstrated in Algorithm 1, we apply SVD on the pre-RoPE key cache and store only the low-rank representations for each layer. Post-RoPE key cache is segmented into chunks, with the mean of each chunk computed as landmarks. By computing the cosine similarity within these chunks, we identify poorly approximated tokens as outliers. This small set of outliers is gathered and stored on the GPU as the static cache, while the remaining key cache is maintained as compact landmarks, with the corresponding values offloaded to the CPU memory.

**High-throughput Decoding.** For incoming queries, we first compute the approximate attention scores using the landmarks. As detailed in Algorithm 2, by identifying the top-k scoring chunk indices, the corresponding values are retrieved from the CPU, and the key cache is simultaneously reconstructed from low-rank projections, effectively concealing the construction of the key cache. Based on the insight that the KV cache has temporal locality, we conduct an index scan to detect the missed chunks and only rebuild the necessary KV pairs on-the-fly, reducing computation and data fetching by 60% with optimized CUDA kernels.

Based on our observations in Section 3.1, future pre-RoPE keys within a sequence reside in a shared low-rank subspace with the context. As a result, an extension of our algorithm would be to store generated tokens as low-rank states using the same low-rank projections obtained from pre-filling to

*Table 1.* Performance comparison of different models and methods on RULER (left side of the table) and LongBench (right side of the table). SHADOWKV outperforms other methods and maintains the accuracy.

| Methods | S1 | S2 | MK1 | MK2 | MQ | MV | QA-1 | QA-2 | VT | FWE | Avg. | NQA | MQA | HQA | MQue | DRead | GRep | SAM | PRetr | LCC | Avg. |
|---|---|---|---|---|---|---|---|---|---|---|---|---|---|---|---|---|---|---|---|---|---|
| *Llama-3-8B-1M* | 100.00 | 100.00 | 98.96 | 98.96 | 98.96 | 95.57 | 75.00 | 48.96 | 78.54 | 71.85 | 86.68 | 18.98 | 41.84 | 36.79 | 21.47 | 31.93 | 34.18 | 35.96 | 81.50 | 56.07 | 39.86 |
| Loki | 18.75 | 1.04 | 2.08 | 0.00 | 1.56 | 0.78 | 4.17 | 13.54 | 26.04 | 25.35 | 9.33 | 2.26 | 10.19 | 5.48 | 3.16 | 12.17 | 28.97 | 7.84 | 40.52 | 31.44 | 15.78 |
| Loki (V) | 41.67 | 6.25 | 37.50 | 1.04 | 8.07 | 30.73 | 10.42 | 19.79 | 51.67 | 37.50 | 24.46 | 3.20 | 21.01 | 12.41 | 3.86 | 17.07 | 31.24 | 16.23 | 52.57 | 38.10 | 21.74 |
| InfiniGen | **100.00** | 98.96 | 84.38 | 53.13 | 63.28 | 54.95 | 65.63 | 48.96 | **81.67** | 50.35 | 70.13 | 14.39 | 31.46 | 33.63 | 17.94 | 26.65 | 27.38 | 21.97 | 74.30 | 38.58 | 31.81 |
| InfiniGen (V) | **100.00** | 98.96 | 96.88 | 76.04 | 81.25 | 77.08 | 67.71 | 50.00 | **81.67** | 53.47 | 78.31 | 17.83 | 36.08 | 35.28 | 19.64 | 28.39 | 29.28 | 28.12 | 74.85 | 45.53 | 35.00 |
| Quest | **100.00** | **100.00** | **98.96** | 77.08 | 97.65 | 93.49 | 60.42 | 50.00 | 77.08 | 65.63 | 82.03 | **20.13** | 36.63 | 35.00 | 18.14 | 24.55 | 27.11 | 35.63 | 79.00 | 53.64 | 36.65 |
| Quest (V) | **100.00** | **100.00** | **98.96** | 85.42 | **97.92** | 95.49 | 70.83 | 46.88 | 78.75 | 65.63 | 83.99 | 17.26 | 39.51 | 36.78 | 18.71 | 26.41 | 29.49 | 35.80 | 79.50 | 60.05 | 38.17 |
| SHADOWKV | **100.00** | **100.00** | 97.92 | **98.96** | 96.88 | **95.83** | 72.92 | 52.08 | 81.67 | 72.57 | **86.88** | 17.17 | 39.73 | 38.29 | 21.08 | 31.77 | 31.62 | 35.87 | 80.00 | 63.93 | 39.94 |
| *GLM-4-9B-1M* | 100.00 | 100.00 | 94.79 | 87.50 | 99.74 | 93.75 | 67.71 | 55.21 | 97.29 | 72.22 | 86.82 | 25.44 | 51.09 | 58.67 | 39.61 | 32.04 | 29.97 | 40.31 | 99.00 | 58.02 | 48.24 |
| Loki | 71.88 | 27.08 | 22.92 | 2.08 | 9.90 | 11.46 | 28.13 | 27.08 | 31.04 | 54.17 | 28.57 | 5.82 | 30.60 | 22.73 | 9.20 | 30.09 | 30.35 | 22.70 | 98.92 | 40.77 | 32.35 |
| Loki (V) | 96.88 | 55.21 | 56.25 | 18.75 | 51.04 | 50.52 | 45.83 | 39.58 | 72.71 | 59.72 | 54.65 | 10.89 | 44.97 | 45.44 | 23.51 | 32.07 | **30.56** | 35.34 | **99.50** | 50.27 | 41.39 |
| InfiniGen | **100.00** | 93.75 | 82.29 |  | 79.43 | 60.16 | 57.29 | 53.13 | 92.71 | 57.29 | 67.60 | 23.67 | 46.31 | 55.69 | 33.91 | 27.49 | 25.44 | 33.48 | 91.83 | 36.96 | 41.64 |
| InfiniGen (V) | **100.00** | 96.88 | 87.50 | 7.29 | 95.31 | 75.26 | 56.25 | 54.17 | 95.63 | 60.76 | 72.91 | 25.63 | 48.44 | 57.23 | 36.94 | 29.77 | 26.67 | 36.64 | 93.58 | 46.69 | 44.62 |
| Quest | **100.00** | 95.83 | 90.62 | 54.17 | 94.01 | 76.30 | 55.21 | 52.08 | 95.83 | 64.58 | 77.86 | 23.81 | 44.53 | 56.41 | 35.49 | 23.54 | 21.73 | 37.39 | 87.00 | 43.80 | 41.52 |
| Quest (V) | **100.00** | 96.88 | 93.75 | 72.92 | 95.83 | 83.07 | 56.25 | 53.13 | 96.88 | 65.97 | 81.47 | 26.00 | 46.32 | 57.54 | 36.42 | 24.58 | 24.52 | 37.71 | 93.50 | 46.52 | 43.68 |
| SHADOWKV | **100.00** | **100.00** | 95.83 | 83.33 | 98.70 | 87.76 | 69.79 | 55.21 | 97.50 | 68.06 | 85.62 | 26.50 | 51.31 | 59.09 | 38.87 | 32.92 | 28.54 | 38.70 | 96.50 | 58.55 | 47.89 |
| *Llama-3.1-8B* | 100.00 | 100.00 | 98.96 | 91.67 | 98.96 | 95.31 | 82.29 | 47.92 | 68.96 | 71.18 | 85.53 | 31.56 | 55.10 | 57.65 | 29.46 | 35.26 | 34.45 | 29.84 | 100.00 | 67.31 | 48.96 |
| Loki | 68.75 | 32.29 | 32.29 | 20.83 | 42.71 | 28.65 | 41.67 | 33.33 | 24.79 | 29.86 | 35.52 | 2.31 | 18.89 | 10.64 | 5.47 | 19.30 | 31.16 | 15.91 | 94.88 | 44.60 | 27.02 |
| Loki (V) | 95.83 | 36.46 | 57.29 | 62.50 | 77.86 | 70.83 | 69.79 | 39.58 | 35.21 | 37.50 | 58.29 | 3.93 | 38.59 | 22.85 | 12.96 | 27.43 | 32.22 | 26.43 | 98.25 | 56.11 | 44.25 |
| InfiniGen | **100.00** | 77.08 | 78.13 | 13.54 | 58.07 | 47.40 | 65.63 | 41.67 | 60.83 | 50.35 | 59.27 | 27.23 | 52.72 | 53.89 | 26.81 | 27.72 | 29.61 | 24.42 | 98.93 | 56.89 | 44.25 |
| InfiniGen (V) | **100.00** | 88.54 | 87.50 | 26.04 | 79.43 | 77.08 | 72.92 | 43.75 | 57.08 | 55.21 | 68.76 | 29.73 | 53.47 | 55.11 | 28.72 | 28.55 | 31.42 | 26.76 | 99.17 | 62.66 | 46.18 |
| Quest | **100.00** | 98.96 | 97.92 | 34.38 | 93.49 | 88.54 | 70.83 | 44.79 | 65.63 | **68.40** | 76.29 | 29.70 | 49.04 | 53.96 | 27.18 | 27.16 | 30.43 | 29.85 | 98.50 | 57.35 | 44.80 |
| Quest (V) | **100.00** | 98.96 | 98.96 | 56.25 | 95.83 | 90.63 | 76.04 | 46.88 | 66.25 | 67.36 | 79.72 | 30.02 | 53.97 | 56.39 | 27.06 | 29.06 | 31.65 | 30.23 | 99.00 | 63.89 | 46.81 |
| SHADOWKV | **100.00** | **100.00** | **100.00** | 83.33 | 97.92 | 92.19 | 81.25 | 48.96 | 67.08 | 64.93 | 83.57 | 30.93 | 55.20 | 57.32 | 29.13 | 31.85 | 32.79 | 30.40 | 99.50 | 66.03 | 48.13 |

reduce the memory usage for the future generations[3]. We evaluate it and include the results in Appendix A.1.

### 4.2. Theoretical Equivalent Bandwidth

The benefit of SHADOWKV in terms of increasing throughput can be analyzed through the concept of equivalent bandwidth. Consider each K or V vector as being $M$ bytes in size, with a sequence length of $S$, a chunk size of $C$, a selected chunk budget of $K$, $O$ outliers, and hit rate $\alpha$. During KV selection, SHADOWKV loads $M \times S/C$ bytes using the GPU memory bandwidth $B_{\text{GPU}}$. For value cache fetching, it loads $M \times K \times C$ bytes using the PCIe bandwidth $B_{\text{PCIe}}$ (Sheng et al., 2023). Since value movement and key cache reconstruction can be overlapped, we do not need to count key cache reconstruction here. Following this, SHADOWKV performs standard attention computation for the top-k chunks and predefined outliers, requiring $2M \times (K + O) \times C$ bytes. The equivalent bandwidth of SHADOWKV is defined as below and the GPU memory savings is detailed in Appendix A.2.

$$\widetilde{B} = \frac{2SB_{\text{GPU}}}{S/C + 2(K+O)C + (1-\alpha)KCB_{\text{GPU}}/B_{\text{PCIe}}}$$

For example, assuming C=8, S=128K, K=256, O=48, $B_{\text{PCIe}}$=31.5 GB/s, and $B_{\text{GPU}}$=2 TB/s for A100, the equivalent bandwidth of SHADOWKV is calculated as 7.2 TB/s,

---

[3]If $\boldsymbol{\Psi} \in \mathbb{R}^{d \times r}$ is the right singular matrix calculated from the SVD of pre-RoPE context keys $\boldsymbol{K} \in \mathbb{R}^{s \times d}$, new pre-RoPE keys $\boldsymbol{K}' \in \mathbb{R}^{s_q \times d}$ can be stored as $\boldsymbol{K}'\boldsymbol{\Psi}$ and projected back up with $\boldsymbol{\Psi}^{\top}$ when needed.

which is 3.6× higher than A100 memory bandwidth. This result indicates that SHADOWKV theoretically achieves a high equivalent bandwidth to accelerate attention computation. System implementation is detailed in Appendix B.1.

## 5. Empirical Evaluation

In this section, we showcase the effectiveness and efficiency of SHADOWKV. Specifically,

- In Section 5.1, we show that SHADOWKV can reduce the GPU memory footprint of the KV cache by over 6× without accuracy degradation on a wide range of models and evaluation benchmarks.

- In Section 5.2, we demonstrate SHADOWKV supports 6× larger batch sizes and increase the inference throughput by up to 3.04× without compromising model quality.

- In Section 5.3, we present ablation studies that validate the effectiveness of each component of SHADOWKV in optimizing GPU memory usage and performance.

### 5.1. Accuracy Evaluation

We demonstrate that SHADOWKV can reduce GPU memory usage of the KV cache by 6× while maintaining accuracy on a range of tasks with a minimal sparse KV budget.

**Setup.** We choose four widely used long-context models for our evaluation: Llama-3-8B-1M (Gradient., 2024), GLM-4-9B-1M (GLM et al., 2024), Llama-3.1-8B (Meta AI, 2024), and Yi-9B-200K (AI et al., 2024). We evalu-

ate our approach on three challenging long-context benchmarks: RULER (Hsieh et al., 2024), LongBench (Bai et al., 2023), and Needle In A Haystack (NIAH) (Kamradt, 2023), covering QA, multi-hop, reasoning, summarization, code completion[4]. We set the chunk size to 8, the rank to 160, and the number of outliers to 48 for SHADOWKV.

**Baselines.** We include three dynamic sparse attention methods as baselines: Quest (Tang et al., 2024b), Loki (Singhania et al., 2024), and InfiniGen (Lee et al., 2024b). For all methods, we retain exact pre-filling and perform dynamic sparse attention during decoding, where the computation cost is set to 1/16 of full attention for selecting sparse KV pairs. We include two variants for each baseline: one where all the KV cache is offloaded, and another where only the value cache is offloaded. The former has similar latency to SHADOWKV but a smaller sparse budget since SHADOWKV only needs to fetch the value cache from the CPU. The latter aligns with the same sparse KV cache budget but significantly increases GPU memory usage. The latter one is marked as "(V)" in the table.

**RULER.** As shown in Table 1, SHADOWKV demonstrates excellent performance on 128K contexts. With a fixed sparse budget of 1.56%, other methods experience performance degradation. In contrast, SHADOWKV is more robust and even outperforms original full attention on certain tasks, such as variable tracking. For complex tasks like multi-document QA or multi-key needle retrieval, other methods suffer from significant performance degradation while SHADOWKV does not.

**LongBench.** On LongBench, we evaluate our method with a range of realistic scenarios, including single-/multi-document QA, document summarization, code completion, information retrieval, etc. We only test on samples longer than 4K and set the sparse KV cache budget to 256 for this benchmark since it has shorter inputs compared to RULER. As shown in Table 1, SHADOWKV outperforms other methods consistently and maintains the performance.

**NIAH.** On NIAH dataset, as shown in Figure 6, SHADOWKV shows the ability to process information at different positions across various context windows, ranging from 16K to 1M tokens. More experiments

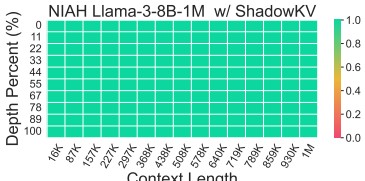

Figure 6. Needle In A Haystack.

on a range of models can be found in Appendix B.3.

**Integrate with Efficient Pre-filling Methods.** We also combined SHADOWKV with a state-of-the-art efficient pre-filling method MInference (Jiang et al., 2024). As shown in Table 2, following the setting of MInference, we tested it on RULER with contexts scaling from 8K to 256K. This demonstrates that our method is compatible with pre-filling acceleration techniques. For some certain context length settings, we even see a slight performance improvement.

Table 2. Performance of different methods on RULER using MInference (Jiang et al., 2024) in the pre-filling stage.

| Methods | 8K | 16K | 32K | 64K | 128K | 256K | Avg. |
|---|---|---|---|---|---|---|---|
| Llama-3-8B-1M | 89.92 | 88.02 | 82.81 | **78.45** | 78.12 | **74.57** | 81.98 |
| SHADOWKV | **90.47** | **88.12** | **83.28** | 77.71 | **78.32** | 74.31 | **82.04** |

**Multi-turn Capability.** To simulate multi-turn conversations, we challenged SHADOWKV with a multi-turn needle retrieval task (Multi-turn NIAH). We also test two eviction-based methods in Figure 7, including SnapKV (Li et al., 2024)

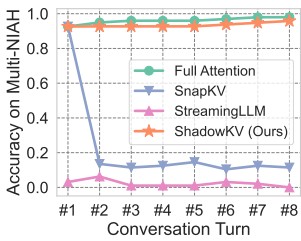

Figure 7. Multi-turn NIAH.

and StreamingLLM (Xiao et al., 2023b). The performance of SnapKV drops significantly from the second round due to the required context information being different from the first round. Since SnapKV inevitably evicted tokens based on the first-turn conversation, it cannot successfully retrieve related information for future queries. In contrast, SHADOWKV can maintain accuracy in the multi-turn conversation setting.

### 5.2. Efficiency Evaluation

To demonstrate the efficiency of SHADOWKV, we deploy it into real-world large batch serving scenarios. By measuring the throughput during decoding across different models on A100, we show that SHADOWKV can support up to 6× larger batch sizes and boost throughput by up to 3.04×.

**Baselines.** The baseline selects the largest batch size that can fit entirely on the GPU with full attention. We also include results for the same batch size of SHADOWKV and the infinite batch size, assuming infinite GPU memory capabilities[5]. We set the sparse budget to 1.56% for SHADOWKV.

---

[4]We include results for Yi-9B-200K and other models (e.g., Llama-3-70B-1M) in Appendix A. Needle In A Haystack is also tested on Phi-3-Mini-128K (Abdin et al., 2024) and Qwen2-7B-128K (Yang et al., 2024a).

[5]For the equivalent SHADOWKV batch size, we evaluate a single Transformer block with FlashAttention and then project the

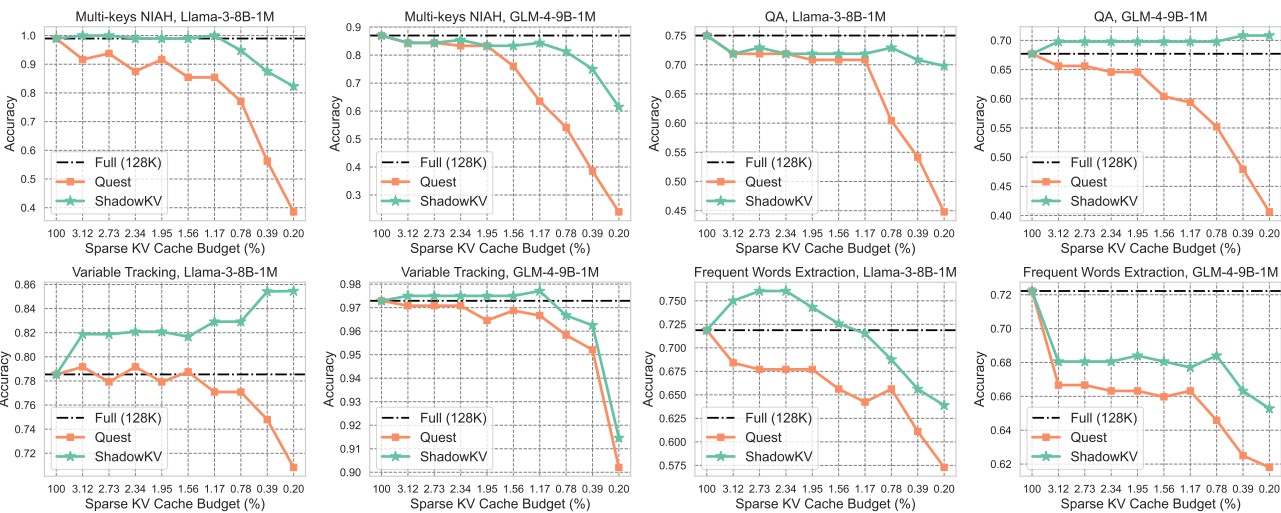

*Figure 8.* Comparison results between different models on a range of long-context tasks with full cache, our SHADOWKV, and Quest. SHADOWKV consistently surpasses Quest under the same sparse budgets and achieves higher throughput.

*Table 3.* Generation throughput (tokens/s) on an A100. The gray text in brackets denotes batch size.

| Model | Context | Full Attn | SHADOWKV | Gain | Full Attn (Inf) |
|---|---|---|---|---|---|
| Llama-3-8B-1M | 60K | 160.62 (8) | **455.14 (48)** | 2.83× | 168.72 (48) / 273.07 (Inf) |
| (8 KV heads) | 122K | 80.77 (4) | **239.51 (24)** | 2.97× | 83.05 (24) / 134.30 (Inf) |
| | 244K | 40.37 (2) | **119.01 (12)** | 2.95× | 52.00 (12) / 67.15 (Inf) |
| Llama-3.1-8B | 60K | 160.93 (8) | **472.77 (48)** | 2.94× | 168.72 (48) / 273.07 (Inf) |
| (8 KV heads) | 122K | 80.78 (4) | **245.90 (24)** | 3.04× | 83.05 (24) / 134.30 (Inf) |
| GLM-4-9B-1M | 60K | 241.05 (12) | **615.89 (50)** | 2.56× | 266.24 (50) / 436.91 (Inf) |
| (4 KV heads) | 122K | 122.67 (6) | **293.40 (25)** | 2.39× | 158.83 (25) / 214.87 (Inf) |
| | 244K | 61.13 (3) | **136.51 (12)** | 2.23× | 78.84 (12) / 107.44 (Inf) |
| Yi-9B-200K | 60K | 204.81 (10) | **544.36 (42)** | 2.66× | 271.21 (42) / 364.09 (Inf) |
| (4 KV heads) | 122K | 101.44 (5) | **260.03 (21)** | 2.56× | 133.53 (21) / 179.06 (Inf) |
| | 244K | 46.74 (2) | **118.55 (10)** | 2.54× | 65.79 (10) / 89.53 (Inf) |

*Table 4.* Generation throughput (tokens/s) under varying batch sizes and sequence lengths on Llama-3-8B-1M.

| Context | 2 | 3 | 4 | 5 | 6 | 8 | 12 | 16 | 24 | 32 | 48 |
|---|---|---|---|---|---|---|---|---|---|---|---|
| **Full KV** | | | | | | | | | | | |
| 60K | 89.19 | 111.44 | 126.73 | 142.62 | 147.40 | 160.62 | OOM | OOM | OOM | OOM | OOM |
| 122K | 65.12 | 75.16 | 80.77 | OOM | OOM | OOM | OOM | OOM | OOM | OOM | OOM |
| 244K | 40.37 | OOM | OOM | OOM | OOM | OOM | OOM | OOM | OOM | OOM | OOM |
| 488K | OOM | OOM | OOM | OOM | OOM | OOM | OOM | OOM | OOM | OOM | OOM |
| **SHADOWKV** | | | | | | | | | | | |
| 60K | 89.69 | 126.61 | 159.41 | 184.92 | 205.41 | 244.20 | 306.09 | 346.34 | 399.65 | 428.63 | **455.14** |
| 122K | 65.61 | 94.28 | 115.01 | 132.23 | 143.77 | 166.72 | 196.73 | 217.24 | **239.51** | OOM | OOM |
| 244K | 48.39 | 65.95 | 78.92 | 87.83 | 94.07 | 104.73 | **119.01** | OOM | OOM | OOM | OOM |
| 488K | 29.82 | 41.01 | 47.13 | 50.85 | **53.46** | OOM | OOM | OOM | OOM | OOM | OOM |

**Results.** As shown in Table 3, SHADOWKV demonstrates significant throughput improvements for various models on an A100, surpassing even those with infinite GPU memory. Notably, SHADOWKV supports batch sizes up to 6× larger and enhances throughput by up to 3.04× compared to full attention, even surpassing infinite batch size assuming infinite GPU memory. While the gains for GLM-4-9B-1M and Yi-9B-200K are slightly lower, the improvements still reach up to 2.56× and 2.66× respectively, highlighting SHADOWKV's adaptability even with fewer KV heads. To further illustrate scalability, we provide detailed throughput measurements for Llama-3-8B-1M under varying batch sizes and context lengths in Table 4. As context length increases, full attention quickly runs out of memory, while SHADOWKV continues to scale up to batch size 48 at 60K context

number to the entire model. For the infinite batch size, we leverage A100's theoretical memory bandwidth (2 TB/s) for attention computations.

and remains functional for extremely longer sequences. Additionally, we provide an efficiency comparison with Quest under 1M contexts in Appendix A.6, demonstrating that SHADOWKV significantly enhances throughput.

### 5.3. Ablation Results

In this section, we present extensive ablation studies of SHADOWKV, focusing on three key points: (1) sparse KV cache budget variations, (2) chunk size selections, (3) pre-RoPE key cache rank choices, (4) latency breakdown, and (5) accuracy contribution of outlier KV Cache. Additional ablations, including precision sensitivity analysis, are provided in Appendix A.

**Sparse KV Cache Budget.** We examine SHADOWKV's performance across various tasks with different sparse budgets, as illustrated in Figure 8. SHADOWKV consistently surpasses Quest under the same sparse budgets and achieves higher throughput. On most tasks, it maintains accuracy

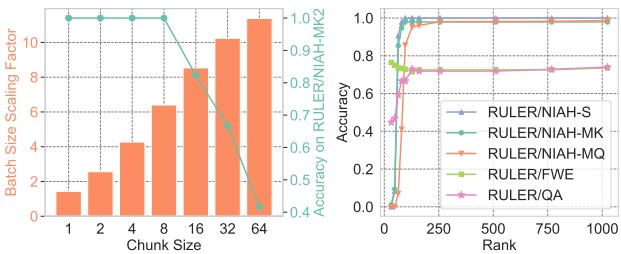

*Figure 9.* **Left:** Impact of chunk size on batch size and accuracy. **Right:** Accuracy trends across different ranks.

*Table 5.* Latency breakdown (ms) of a Transformer block of Llama-3-8B-1M during prefilling.

| Context | Attention | FFN | SVD | Reduce | CosineSimilarity | TopK | Gather | Cost |
|---|---|---|---|---|---|---|---|---|
| 64K | 186.23 | 96.47 | 17.19 | 0.10 | 1.41 | 0.08 | 0.01 | 6.65% |
| 128K | 721.13 | 193.32 | 26.62 | 0.20 | 2.77 | 0.14 | 0.02 | 3.25% |
| 256K | 2880.12 | 392.77 | 50.56 | 0.42 | 6.11 | 0.11 | 0.03 | 1.75% |
| 512K | 11720.30 | 789.23 | 108.38 | 0.84 | 12.19 | 0.15 | 0.06 | 0.97% |

with just a 1.56% sparse budget compared to full attention and even improves slightly on some tasks.

**Chunk Size.** As shown in Figure 9, increasing the chunk size allows for larger batch sizes. However, accuracy declines when the chunk size exceeds eight. Meanwhile, the chunk size choice has minimal impact on the chunk hit rate, which remains around 60%.

**Rank of Pre-RoPE Keys.** We assess SHADOWKV's performance across various tasks using different ranks for pre-RoPE keys. As illustrated in Figure 9, accuracy increases with the rank up to approximately 160, after which it stabilizes near full-rank performance. Interestingly, the trends vary across tasks, and in some cases, low-rank approximations achieve better performance.

**Scalability for Longer Sequences.** As shown in Table 5, the overhead of SVD, reduce, cosine similarity, topK, and gather computing is very low and tends to decrease as the sequence scales, proving that SHADOWKV's scalability to longer sequences.

**Overlapping Operations for Latency Reduction.** In Table 6, we demonstrate how overlapping the recomputation of the key cache with value cache fetching from the CPU significantly reduces decoding latency. This concurrent processing approach ensures that SHADOWKV minimizes overhead when handling long-context models.

**Accuracy Contribution of Outlier KV Cache.** We conduct experiments using different numbers of outlier chunks

*Table 6.* Latency breakdown (ms) of a Transformer block of Llama-3-8B-1M during decoding.

| Context | GEMM+ Softmax | Max | TopK | Recompute K (Overlapped) | Fetch V | Attention | FFN | QKV |
|---|---|---|---|---|---|---|---|---|
| 48×64K | 0.56 | 0.07 | 0.14 | 1.25 | 1.84 | 0.23 | 0.33 | 0.05 |
| 24×128K | 0.58 | 0.07 | 0.15 | 1.36 | 1.66 | 0.21 | 0.29 | 0.05 |
| 12×256K | 0.65 | 0.07 | 0.16 | 1.49 | 1.75 | 0.19 | 0.25 | 0.05 |
| 6×512K | 0.71 | 0.07 | 0.17 | 1.51 | 1.69 | 0.18 | 0.23 | 0.05 |

*Table 7.* Performance across different number of outlier chunks on RULER (Hsieh et al., 2024) evaluated at length of 128K.

| # Outliers | N-S1 | N-S2 | N-MK1 | N-MK2 | N-MQ | N-MV | QA-1 | QA-2 | VT | FWE | Avg. |
|---|---|---|---|---|---|---|---|---|---|---|---|
| 0 (0.000 %) | 100.00 | 100.00 | 96.88 | 85.42 | 73.18 | 70.83 | 43.75 | 39.58 | 73.54 | 57.29 | 74.05 |
| 1 (0.006 %) | 100.00 | 100.00 | 97.92 | 98.96 | 95.83 | 94.79 | 70.83 | 51.04 | 70.63 | 70.14 | 85.01 |
| 2 (0.012 %) | 100.00 | 100.00 | 97.92 | 98.96 | 95.57 | 95.57 | 70.83 | 51.04 | 72.08 | 70.49 | 85.25 |
| 4 (0.024 %) | 100.00 | 100.00 | 97.92 | 98.96 | 95.83 | 95.57 | 71.88 | 51.04 | 74.38 | 71.18 | 85.68 |
| 8 (0.049 %) | 100.00 | 100.00 | 97.92 | 98.96 | 95.57 | 95.05 | 72.92 | 51.04 | 78.13 | 72.57 | 86.22 |
| 16 (0.098 %) | 100.00 | 100.00 | 97.92 | 98.96 | 96.09 | 95.31 | 72.92 | 51.04 | 80.42 | 71.53 | 86.42 |
| 32 (0.195 %) | 100.00 | 100.00 | 97.92 | 98.96 | 96.35 | 95.57 | 72.92 | 52.08 | 81.25 | 72.22 | 86.73 |
| 48 (0.293 %) | 100.00 | 100.00 | 97.92 | 98.96 | 96.88 | 95.83 | 72.92 | 52.08 | 81.67 | 72.57 | 86.88 |
| *Quest (Ref.)* | 100.00 | 100.00 | 98.96 | 77.08 | 97.65 | 93.49 | 60.42 | 50.00 | 77.08 | 65.63 | 82.03 |
| *Full Attn (Ref.)* | 100.00 | 100.00 | 98.96 | 98.96 | 98.96 | 95.57 | 75.00 | 48.96 | 78.54 | 71.85 | 86.68 |

for Llama-3-8B-1M on the RULER benchmark with 128K context length. As presented in Table 7, our findings indicate that outliers play a crucial role. For instance, the first chunk, a significant outlier, has previously been shown to act as an attention sink (Xiao et al., 2023b), underscoring its importance in maintaining model accuracy. Remarkably, with just 8 outliers (0.049%), SHADOWKV outperforms the Quest baseline and nearly matches the accuracy achieved by full attention. However, when outliers are not adequately managed, the performance of the mean-based landmarks in SHADOWKV may fall below the min-max approach used by Quest, underscoring the importance of handling outliers properly.

## 6. Conclusion

We present SHADOWKV, a high-throughput inference system for long-context LLM inference. SHADOWKV optimizes GPU memory usage through the low-rank key cache and offloaded value cache, allowing for larger batch sizes. It reduces decoding overhead by accurate sparse attention, boosting throughput while maintaining accuracy. Our empirical experiments demonstrate SHADOWKV can support up to 6× larger batch sizes and enhance throughput by up to 3.04× on an A100 across various long-context models, including Llama-3.1-8B, Llama-3-8B-1M, GLM-4-9B-1M, and Yi-9B-200K. SHADOWKV holds great promise for improving long-context LLM inference.

## Acknowledgment

We thank Zhuoming Chen, Xinyu Yang, and Yang Zhou for their helpful discussions and feedback on early drafts of the paper. This work was partially supported by Li-Auto, Amazon, Intel, Moffet AI, Cylab seed.

## Impact Statement

The deployment of long-context LLMs is often constrained by their significant memory and computational demands, limiting accessibility for real-world applications that require handling extended contexts. This work introduces SHADOWKV, a high-throughput inference system that addresses these constraints by optimizing the KV cache. By leveraging low-rank key compression and efficient value offloading, SHADOWKV reduces GPU memory usage and inference latency while supporting larger batch sizes and longer sequences. These innovations enable more efficient use of existing hardware, reducing the resource barriers associated with deploying LLMs.

The societal implications of SHADOWKV are multifaceted. By making it feasible to deploy LLMs on constrained hardware, this work democratizes access to advanced AI capabilities, empowering smaller organizations and researchers in resource-limited settings. Furthermore, its ability to maintain accuracy while reducing computational overhead contributes to the broader sustainability goals of AI by lowering energy consumption in large-scale deployments.

No specific ethical concerns or societal risks are associated with the proposed method. Instead, it provides a path toward equitable and responsible AI deployment, ensuring that advancements in LLM capabilities are accessible and sustainable across diverse sectors.

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

# A. Additional Experiment Results

In this section, we present additional experiments experiments not covered in the main text, including the handling of newly generated tokens (as discussed in Section 4.1), scalability analysis for larger models and longer sequences (mentioned in Section 5.1), additional ablation studies (referenced in Section 5.3), and etc.

## A.1. Handling of Newly Generated Tokens

To address the handling of newly generated tokens, we project these tokens' key cache into a low-rank space using the same projections applied during the prefilling phase. This approach preserves the benefits of reduced GPU memory usage, particularly for long output sequences.

As shown in Table 8 and Table 9, we refer to this extension as SHADOWKV+. Our evaluation across various models demonstrates that SHADOWKV effectively maintains accuracy while optimizing memory usage.

*Table 8.* Performance of SHADOWKV and SHADOWKV+ across different models on RULER (Hsieh et al., 2024) evaluated at length of 128K.

| Methods | N-S1 | N-S2 | N-MK1 | N-MK2 | N-MQ | N-MV | QA-1 | QA-2 | VT | FWE | Avg. |
|---|---|---|---|---|---|---|---|---|---|---|---|
| *Llama-3-8B-1M* | 100.00 | 100.00 | 98.96 | 98.96 | 98.96 | 95.57 | 75.00 | 48.96 | 78.54 | 71.85 | 86.68 |
| SHADOWKV | **100.00** | **100.00** | 97.92 | 98.96 | 96.88 | **95.83** | **72.92** | **52.08** | **81.67** | **72.57** | **86.88** |
| SHADOWKV+ | **100.00** | **100.00** | **98.96** | **100.00** | 95.83 | 93.49 | 71.88 | 50.00 | 80.21 | 71.88 | 86.23 |
| *GLM-4-9B-1M* | 100.00 | 100.00 | 94.79 | 87.50 | 99.74 | 93.75 | 67.71 | 55.21 | 97.29 | 72.22 | 86.82 |
| SHADOWKV | **100.00** | **100.00** | **95.83** | 83.33 | **98.70** | **87.76** | **69.79** | 55.21 | 97.50 | **68.06** | 85.62 |
| SHADOWKV+ | **100.00** | **100.00** | **95.83** | 85.42 | 98.17 | 85.16 | **69.79** | **56.25** | **97.92** | 67.71 | **85.63** |
| *Llama-3.1-8B* | 100.00 | 100.00 | 98.96 | 91.67 | 98.96 | 95.31 | 82.29 | 47.92 | 68.96 | 71.18 | 85.53 |
| SHADOWKV | **100.00** | **100.00** | **100.00** | 83.33 | **97.92** | **92.19** | **81.25** | 48.96 | **67.08** | **64.93** | 83.57 |
| SHADOWKV+ | **100.00** | **100.00** | **100.00** | 84.38 | 96.88 | 91.67 | **81.25** | **52.08** | 65.63 | 62.85 | 83.47 |
| *Yi-9B-200K* | 100.00 | 100.00 | 86.46 | 62.50 | 64.58 | 32.55 | 44.79 | 39.58 | 36.87 | 89.93 | 65.73 |
| SHADOWKV | **100.00** | **100.00** | **82.29** | **67.71** | **63.28** | 31.51 | 43.75 | **38.54** | **56.04** | 72.22 | **65.53** |
| SHADOWKV+ | **100.00** | **100.00** | 81.25 | **67.71** | 61.72 | 31.51 | **46.88** | **38.54** | 53.54 | **72.92** | 65.41 |

*Table 9.* Performance of SHADOWKV and SHADOWKV+ on different models with LongBench (Bai et al., 2023) samples exceeding 4K tokens.

| Methods | NarratQA | MultiFQA | HotpotQA | MuSiQue | DuRead | GovRep | SAMSum | PassRetr | LCC | Avg. |
|---|---|---|---|---|---|---|---|---|---|---|
| *Llama-3-8B-1M* | 18.98 | 41.84 | 36.79 | 21.47 | 31.93 | 34.18 | 35.96 | 81.50 | 56.07 | 39.86 |
| SHADOWKV | 17.17 | 39.73 | **38.29** | **21.08** | **31.77** | 31.62 | **35.87** | **80.00** | **63.93** | 39.94 |
| SHADOWKV+ | **20.42** | **41.16** | 37.22 | 21.03 | **31.77** | **31.98** | 35.80 | **80.00** | 63.89 | **40.36** |
| *GLM-4-9B-1M* | 25.44 | 51.09 | 58.67 | 39.61 | 32.04 | 29.97 | 40.31 | 99.00 | 58.02 | 48.24 |
| SHADOWKV | 26.50 | **51.31** | 59.09 | **38.87** | 32.92 | 28.54 | 38.70 | **96.50** | **58.55** | 47.89 |
| SHADOWKV+ | **27.59** | **51.31** | **59.17** | 38.34 | **33.55** | **31.25** | **39.46** | **96.50** | 55.86 | **48.11** |
| *Llama-3.1-8B* | 31.56 | 55.10 | 57.65 | 29.46 | 35.26 | 34.45 | 29.84 | 100.00 | 67.31 | 48.96 |
| SHADOWKV | 30.93 | **55.20** | 57.32 | **29.13** | 31.85 | 32.79 | **30.40** | **99.50** | 66.03 | **48.13** |
| SHADOWKV+ | **32.25** | 54.29 | **57.75** | 28.37 | 31.07 | **32.89** | 28.73 | 98.75 | **67.59** | 47.97 |
| *Yi-9B-200K* | 13.88 | 30.02 | 52.46 | 28.20 | 22.29 | 30.25 | 19.08 | 67.00 | 73.50 | 37.41 |
| SHADOWKV | 12.44 | 30.82 | **52.43** | **27.73** | **20.79** | **29.83** | 20.73 | 64.00 | 72.89 | 36.85 |
| SHADOWKV+ | **14.08** | **30.94** | 51.16 | 27.00 | 19.50 | 29.34 | **21.16** | **66.00** | **73.47** | **36.96** |

## A.2. Quantitative Analysis of GPU Memory Savings

The GPU memory savings provided by SHADOWKV can be quantitatively analyzed as follows. Let each K or V vector have a size of $M$ bytes, with a sequence length $S$, a chunk size $C$, a selected chunk budget $K$, $O$ outliers, and a pre-RoPE key cache rank $r$. The GPU memory savings of SHADOWKV can then be expressed as:

$$\text{Memory Savings} = \frac{2SM}{SM/C + 2(K+O)C + Sr + rM}$$

For example, assuming $M = 1024, C = 8, S = 128\text{K}, K = 256, O = 48, r = 160$, the memory savings of SHADOWKV is calculated as $7.08\times$. This result demonstrates that SHADOWKV can theoretically reduce the KV cache memory footprint on the GPU by $7.08\times$ for longer sequences and larger batch sizes.

## A.3. Accuracy Results for Yi-9B-200K

We present accuracy results for Yi-9B-200K (AI et al., 2024) on RULER (Hsieh et al., 2024) and LongBench (Bai et al., 2023), highlighting SHADOWKV's superior performance across diverse tasks compared to other methods.

Table 10. Performance of Yi-9B-200K with different methods on RULER (Hsieh et al., 2024) evaluated at length of 128K. SHADOWKV outperforms other methods with a 1.56% sparse budget.

| Methods | N-S1 | N-S2 | N-MK1 | N-MK2 | N-MQ | N-MV | QA-1 | QA-2 | VT | FWE | Avg. |
|---|---|---|---|---|---|---|---|---|---|---|---|
| *Yi-9B-200K* | 100.00 | 100.00 | 86.46 | 62.50 | 64.58 | 32.55 | 44.79 | 39.58 | 36.87 | 89.93 | 65.73 |
| Loki | 34.38 | 2.08 | 2.08 | 0.00 | 0.00 | 0.52 | 22.92 | 21.88 | 0.00 | 25.00 | 10.89 |
| Loki (V only) | 59.38 | 11.46 | 18.75 | 5.21 | 4.43 | 2.08 | 22.92 | 31.25 | 0.00 | 35.07 | 19.06 |
| InfiniGen | **100.00** | 94.79 | 77.08 | 1.04 | 40.10 | 20.57 | 37.50 | 34.38 | 41.46 | 46.18 | 49.31 |
| InfiniGen (V only) | **100.00** | 98.96 | 78.13 | 2.08 | 58.33 | 24.48 | 40.63 | 35.42 | 52.92 | 55.90 | 54.69 |
| Quest | **100.00** | 98.96 | 79.17 | 26.04 | 56.51 | 31.77 | 32.29 | 31.25 | 51.04 | 71.88 | 57.89 |
| Quest (V only) | **100.00** | 100.00 | 80.21 | 45.83 | 59.37 | **31.90** | 36.45 | 34.37 | 53.54 | 71.88 | 61.36 |
| SHADOWKV | **100.00** | **100.00** | **82.29** | **67.71** | **63.28** | 31.51 | **43.75** | **38.54** | **56.04** | **72.22** | **65.53** |

Table 11. Performance of Yi-9B-200K with LongBench (Bai et al., 2023) samples exceeding 4K tokens. SHADOWKV outperforms other methods and maintains the accuracy.

| Methods | NarrQA | MultiFQA | HotpotQA | MuSiQue | DuRead | GovRep | SAMSum | PassRetr | LCC | Avg. |
|---|---|---|---|---|---|---|---|---|---|---|
| *Yi-9B-200K* | 13.88 | 30.02 | 52.46 | 28.20 | 22.29 | 30.25 | 19.08 | 67.00 | 73.50 | 37.41 |
| Loki | 1.63 | 2.73 | 16.21 | 4.87 | 4.75 | 2.13 | 4.95 | 0.00 | 38.72 | 8.44 |
| Loki (V only) | 1.96 | 10.39 | 21.31 | 7.36 | 6.78 | 9.15 | 10.02 | 4.00 | 58.75 | 14.41 |
| InfiniGen | 10.01 | 23.61 | 50.47 | 25.91 | 15.11 | 27.96 | 18.97 | 30.00 | 56.46 | 28.72 |
| InfiniGen (V only) | 11.31 | 26.46 | 51.13 | 26.77 | 16.09 | 28.67 | 19.33 | 34.00 | 62.07 | 30.65 |
| Quest | 10.57 | 25.83 | 46.06 | 23.04 | 17.09 | 17.11 | 20.59 | 50.50 | 67.70 | 30.94 |
| Quest (V only) | **14.56** | 25.73 | 48.73 | 24.73 | 18.44 | 20.83 | 20.08 | 57.50 | 71.13 | 33.53 |
| SHADOWKV | 12.44 | **30.82** | **52.43** | **27.73** | **20.79** | **29.83** | **20.73** | **64.00** | **72.89** | **36.85** |

## A.4. Precision Sensitivity

In the main experiments, we used BF16 for both model weights and KV cache. To further investigate the impact of precision on SHADOWKV's performance, we conducted additional experiments using FP8 precision (`torch.float8_e5m2`). These tests aim to determine whether SHADOWKV can retain its accuracy at this lower precision, addressing concerns about precision sensitivity, particularly in SVD computations.

As detailed in Table 12 and Table 13, SHADOWKV and baseline methods were evaluated using FP8. Results show that SHADOWKV maintains accuracy and achieves consistently high performance even with FP8 precision. This robustness, despite FP8's reduced numerical range, confirms that SHADOWKV can continue to deliver efficiency gains without compromising accuracy.

*Table 12.* Performance comparison of SHADOWKV and baseline methods on the RULER (Hsieh et al., 2024) using FP8 precision, evaluated at a sequence length of 128K.

| Methods | N-S1 | N-S2 | N-MK1 | N-MK2 | N-MQ | N-MV | QA-1 | QA-2 | VT | FWE | Avg. |
|---|---|---|---|---|---|---|---|---|---|---|---|
| *Llama-3-8B-1M* | 100.00 | 100.00 | 98.96 | 95.83 | 97.40 | 95.57 | 63.54 | 48.96 | 75.83 | 73.26 | 84.94 |
| Loki | 5.21 | 1.04 | 0.00 | 0.00 | 0.78 | 0.26 | 5.21 | 13.54 | 28.33 | 28.82 | 8.32 |
| Loki (V only) | 36.46 | 9.38 | 31.25 | 0.00 | 6.25 | 21.09 | 11.46 | 15.63 | 57.08 | 35.76 | 22.44 |
| Quest | **100.00** | 98.96 | **98.96** | 71.88 | 96.61 | **93.49** | 63.54 | 45.83 | 78.13 | 67.01 | 81.44 |
| Quest (V only) | **100.00** | **100.00** | **98.96** | 85.42 | **97.40** | **93.49** | 70.83 | **48.96** | 78.13 | 65.63 | 83.88 |
| SHADOWKV | **100.00** | **100.00** | 97.92 | **94.79** | 95.31 | **93.49** | **75.00** | **48.96** | **80.42** | **73.61** | **85.95** |

*Table 13.* Evaluation of SHADOWKV and baseline methods on LongBench (Bai et al., 2023) with sequence lengths exceeding 4K tokens, using FP8 precision.

| Methods | NarratQA | MultiFQA | HotpotQA | MuSiQue | DuRead | GovRep | SAMSum | PassRetr | LCC | Avg. |
|---|---|---|---|---|---|---|---|---|---|---|
| *Llama-3-8B-1M* | 18.69 | 41.21 | 35.76 | 21.59 | 31.81 | 33.77 | 35.29 | 80.50 | 56.77 | 39.49 |
| Loki | 2.21 | 11.12 | 5.70 | 1.84 | 15.42 | 28.59 | 11.41 | 41.91 | 33.99 | 16.91 |
| Loki (V only) | 2.68 | 22.33 | 12.69 | 3.35 | 21.43 | 30.57 | 16.32 | 47.68 | 36.64 | 21.52 |
| Quest | 19.41 | 38.92 | 34.02 | 19.64 | 23.13 | 26.40 | 28.04 | 78.50 | 49.81 | 35.32 |
| Quest (V only) | 16.19 | 36.73 | **36.64** | 19.59 | 25.57 | 29.46 | 27.14 | **79.50** | 60.05 | 36.76 |
| SHADOWKV | **18.29** | **39.39** | 36.06 | **21.04** | 30.47 | 31.87 | **35.56** | 78.50 | **62.11** | **39.25** |

## A.5. Scalability Analysis for Larger Models and Longer Sequences

To demonstrate the scalability of SHADOWKV, we present experiments with Llama-3-8B-1M on 1M contexts and Llama-3-70B-1M on 512K contexts, using the RULER benchmark (Hsieh et al., 2024). Additionally, we evaluate Llama-3-70B-1M on the Needle In A Haystack dataset, testing context lengths ranging from 16K to 1M tokens.

As shown in Figure 10 and Table 14, SHADOWKV maintains robust performance across increasing context lengths and model sizes, demonstrating its scalability in handling large-scale inputs. This scalability allows SHADOWKV to process extensive contexts with high accuracy, making it a valuable solution for real-world applications requiring extensive sequences.

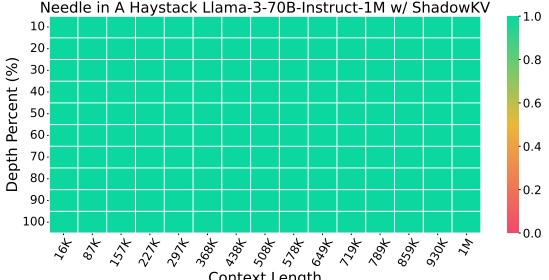

*Figure 10.* Needle In A Haystack.

*Table 14.* Performance of different methods on RULER (Hsieh et al., 2024) evaluated at length of 1M. The Llama-3-8B-1M is evaluated on 1M contexts while the Llama-3-70B-1M is evaluated on 512K contexts.

| Methods | N-S1 | N-S2 | N-MK1 | N-MK2 | N-MQ | N-MV | QA-1 | QA-2 | VT | FWE | Avg. |
|---|---|---|---|---|---|---|---|---|---|---|---|
| *Llama-3-70B-1M* | 100.00 | 82.29 | 90.63 | 54.17 | 85.16 | 96.61 | 69.79 | 35.42 | 68.75 | 69.44 | 75.23 |
| Loki | 100.00 | 1.04 | 0.00 | 0.00 | 0.00 | 0.00 | 13.54 | 11.46 | 34.30 | 22.92 | 18.33 |
| Loki (V only) | 100.00 | 15.63 | 26.04 | 0.00 | 0.00 | 0.00 | 25.00 | 19.79 | 40.00 | 31.94 | 25.84 |
| Quest | 100.00 | 76.04 | 78.13 | 35.42 | 85.47 | 92.19 | 53.21 | 34.38 | 38.33 | 58.33 | 65.15 |
| Quest (V only) | 100.00 | 77.08 | 79.17 | 36.49 | 86.19 | **95.31** | 54.17 | 36.58 | 47.70 | 58.68 | 67.14 |
| SHADOWKV | **100.00** | **82.29** | **88.54** | **53.04** | **88.02** | 94.79 | **67.71** | **37.50** | **68.54** | **68.25** | **74.87** |
| *Llama-3-8B-1M* | 96.88 | 100.00 | 96.88 | 69.79 | 91.15 | 85.68 | 64.58 | 42.71 | 25.00 | 56.25 | 72.89 |
| Loki | 9.38 | 1.04 | 10.42 | 0.00 | 2.60 | 4.43 | 38.54 | 11.46 | 1.67 | 0.69 | 8.02 |
| Loki (V only) | 68.75 | 29.17 | 60.42 | 1.04 | 26.56 | 43.23 | 59.38 | 15.63 | 6.46 | 0.69 | 31.13 |
| Quest | 94.79 | 92.71 | 80.21 | 4.17 | 76.30 | 69.27 | 57.29 | 28.13 | 25.67 | 30.56 | 55.91 |
| Quest (V only) | 94.79 | 93.75 | 81.25 | 4.17 | 79.69 | 69.27 | 62.50 | 31.25 | 26.00 | 32.99 | 57.57 |
| SHADOWKV | **96.88** | **100.00** | **96.88** | **65.63** | **89.38** | **83.16** | **69.79** | **42.71** | **26.04** | **59.38** | **72.98** |

### A.6. Efficiency Comparison with Quest

We present an efficiency comparison with Quest, particularly under long contexts or high batch sizes where the GPU memory alone cannot accommodate the KV cache. In such cases, both Full Attention and Quest must offload the KV cache to the CPU. As shown in Table 15, SHADOWKV significantly outperforms both Full Attention and Quest under the same sparse budget.

The efficiency advantage of SHADOWKV over Quest is due to two key factors: (1) SHADOWKV only fetches the value cache from the CPU, rather than the entire KV pair, minimizing data transfer and reducing latency, and (2) SHADOWKV integrates a cache mechanism that leverages the temporal locality of the KV cache.

*Table 15.* Efficiency comparsion with Quest.

| Context | Full Attention | Full Attention (CPU) | Quest | Quest (CPU) | SHADOWKV |
|---------|----------------|----------------------|-------|-------------|----------|
| 3×1M | OOM | 0.21 tokens/s | OOM | 9.34 tokens/s | 45.32 tokens/s |

### A.7. Detailed Comparison with InfiniGen

We provide further clarification on the key distinctions and conduct additional experiments between SHADOWKV and InfiniGen. These experiments show that SHADOWKV significantly outperforms InfiniGen across a wide range of downstream tasks.

**Differences in SVD Usage.** Infinigen performs an offline SVD to get a projection matrix, which is applied to post-RoPE key and query states for KV selection, while SHADOWKV applies an online, prompt-dependent SVD directly to the pre-RoPE key cache for compression, not for KV selection.

**Methodological Differences.** While InfiniGen uses SVD for KV selection, it requires fetching selected, exact KV pairs from the CPU. In contrast, SHADOWKV only fetches the value cache from the CPU, reconstructing the key cache from its low-rank storage on the GPU. By overlapping these processes, SHADOWKV reduces data-fetch overhead and achieves improved efficiency in KV cache management.

**Accuracy Comparison.** To empirically validate SHADOWKV's advantages, we conducted accuracy evaluations. Results confirm SHADOWKV's effectiveness in maintaining accuracy while optimizing memory usage. Although InfiniGen performs well on simpler tasks like RULER-N-S1, it shows significant accuracy drops on more complex tasks, such as RULER-N-MK2, RULER-FWE, LongBench-LCC, and others, where SHADOWKV maintains consistently high accuracy.

# B. Experiment Details

In this section, our goal is to provide the details of the system implementation (mentioned in Section 4.2), experiment settings, and additional experiments on InfiniteBench (Zhang et al., 2024b) and Needle In A Haystack (Kamradt, 2023).

## B.1. System Implementation.

We implement the framework based on PyTorch (Paszke et al., 2019; Wolf, 2019) and dedicated kernels (Thakkar et al., 2023). FlashAttention (Dao et al., 2022; Dao, 2023; Hong et al., 2023) is used for attention computation and some efficient fused kernels in Flashinfer (Ye et al., 2024) and vLLM (Kwon et al., 2023) are used, including layer norm. To reduce memory movement and kernel launch overhead, we fuse some operations into CUDA kernels, including attention approximation, key cache low-rank reconstruction, value cache fetching, cache mechanism, etc. We leverage multi-streams to overlap the reconstruction of key cache and value cache fetching. We set the rank of pre-RoPE key cache to 160, chunk size to 8, and sparse KV cache budget to 1.56% for most cases.

## B.2. Dataset Details

LLMs are widely used in various fields (Sun et al., 2024b; Peng et al., 2025; Fu et al., 2025; Li et al., 2023; Yuan et al., 2024; QwenTeam, 2024; Wang et al., 2024a; Song et al., 2024), and we select three long-context benchmarks, detailed below.

- RULER (Hsieh et al., 2024) consists of 13 complex tasks and supports adjustable context lengths, including retrieval, multi-hop tracking, aggregation, and QA tasks. For the test with MInference (Jiang et al., 2024), we set up test sets scaling from 8K to 256K for evaluation.

- LongBench (Bai et al., 2023) is a challenging long-context benchmark that assesses the performance of LLMs in extended contexts. Featuring Chinese and English languages, LongBench encompasses 6 main categories and 21 diverse tasks, evaluating LLM capabilities across crucial long-text applications like single-/multi-document QA, summarization, code completion, etc.

- Needle In A Haystack (Kamradt, 2023) is a long-context retrieval benchmark testing LLM's performance with context window scales up to 1M tokens where information placed at various positions. We tested the retrieval capabilities of six long-context LLMs based on their context length.

## B.3. Needle In A Haystack

In addition to the Needle In A Haystack results for Llama-3-8B-1M shown in Figure 6, we also present results for GLM-4-9B-1M, Llama-3.1-8B, Yi-9B-200K, Phi-3-Mini-128K, and Qwen2-7B-128K, shown in Figure 11. Compared to full attention, using SHADOWKV has minimal impact on the ability to understand semantic information across different context windows and needle depths. There is even a slight performance improvement for Yi-9B-200K.

## B.4. InfiniteBench

InfiniteBench (Zhang et al., 2024b) is a challenging long-context benchmark that consists of 10 tasks, including QA, coding, dialogue, summarization, and retrieval, with an average length of 214K.

*Table 16.* Accuracy of different methods on InfiniteBench (Zhang et al., 2024b).

| Methods | En.Sum | En.QA | En.MC | En.Dia | Zh.QA | Code.Debug | Math.Find | Retr.PassKey | Retr.Num |
|---|---|---|---|---|---|---|---|---|---|
| *Llama-3-8B-1M* | 23.05 | 18.14 | 65.06 | 10.50 | 12.47 | 24.36 | 37.14 | 100.00 | 100.00 |
| SHADOWKV | 21.50 | 17.73 | 64.63 | 10.50 | 12.45 | 23.86 | 37.43 | 100.00 | 100.00 |
| *GLM-4-9B-1M* | 28.61 | 9.25 | 68.12 | 39.50 | 11.77 | 30.20 | 40.00 | 100.00 | 100.00 |
| SHADOWKV | 23.22 | 8.48 | 68.56 | 32.50 | 11.27 | 30.46 | 40.00 | 100.00 | 100.00 |
| *Llama-3.1-8B* | 26.42 | 14.48 | 66.38 | 16.00 | 12.92 | 21.07 | 34.00 | 100.00 | 99.66 |
| SHADOWKV | 24.23 | 13.83 | 66.38 | 16.50 | 12.76 | 21.07 | 34.00 | 100.00 | 94.41 |
| *Yi-9B-200K* | 8.88 | 10.61 | 61.57 | 5.50 | 13.88 | 21.57 | 23.71 | 100.00 | 99.66 |
| SHADOWKV | 8.92 | 10.06 | 59.39 | 6.00 | 13.89 | 20.56 | 24.29 | 100.00 | 99.83 |

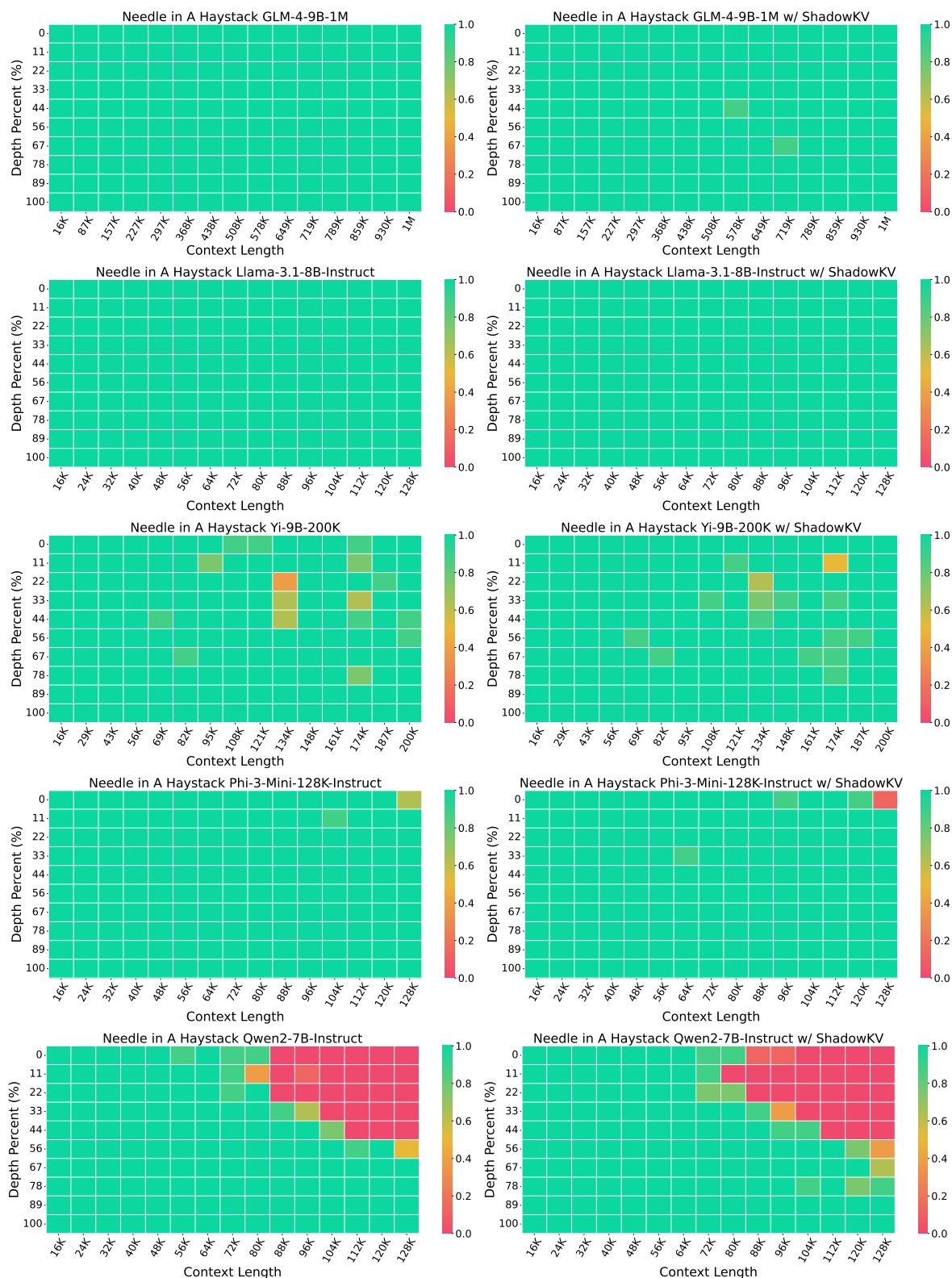

*Figure 11.* Needle In A Haystack ([Kamradt, 2023](#)) results using GLM-4-9B-1M ([GLM et al., 2024](#)), Llama-3.1-8B-Instruct ([Meta AI, 2024](#)), Yi-9B-200K ([AI et al., 2024](#)), Phi-3-Mini-128K ([Abdin et al., 2024](#)), and Qwen2-7B-128K ([Yang et al., 2024a](#)).

