# OpenReview forum: "ShadowKV: KV Cache in Shadows for High-Throughput Long-Context LLM Inference"
_ICML.cc/2025/Conference — ICML 2025 spotlightposter_

### Official Review · Reviewer_1RzE · 2025-03-09

**Overall Recommendation:** 4

**Summary:**

This paper presents ShadowKV, a system for long-context LLM inference that optimizes memory usage and throughput with negligible impact on output quality.

ShadowKV consists of two key techniques:
1. In GPU memory, it only stores the SVD decomposition of pre-ROPE key caches to reduce memory usage per request. The value cache will be stored in CPU memory and be loaded back to GPU during the inference.
2. During the inference, it selects a subset of the tokens' KV cache to run the attention, which further increases the batch size and the inference throughput.

The evaluation results show that Shadow KV achieves similar quality to the SOTA KV cache eviction baselines and can improve the inference throughput compared to full KV cache computation.

## update after rebuttal

Thanks to the authors for answering my questions. The rebuttal has addressed my concerns and I will keep my score of "accept".

**Claims And Evidence:**

The first key claim in this paper is that the Pre-ROPE key tensor is much sparser than other intermediate catchable results during LLM inference. The author provides clear and easy-to-understand evidence in the form of figures to support this claim.

The second key claim in the paper is that the attention scores of the adjacent tokens are usually very similar (with a small amount of outliers). The authors also used real experimental results to support the claim.

However, the authors did not clarify which dataset they used to produce the experimental result, whether the dataset is real or synthetic, and whether it matches the target use case of this paper. This limits the credibility of the claims.

**Essential References Not Discussed:**

N/A

**Experimental Designs Or Analyses:**

The experimental design is clear and comprehensive.

**Methods And Evaluation Criteria:**

The evaluation section consists of 3 main parts:

1. evaluating the impact on the LLM generation quality.
2. evaluating the improvement of the runtime efficiency.
3. evaluating the impact of different configurations in the system

In all three parts, the baselines and the setup are clear and easy to understand. The selected datasets also fit the target use case of this paper.

Overall, the evaluation of the paper is solid.

**Other Comments Or Suggestions:**

I really enjoyed reading this paper and liked the idea of SVD decomposition of pre-ROPE key cache.

**Other Strengths And Weaknesses:**

Strengths
- Well-designed system with clear technical details, covering most aspects during the inference.
- Solid experimental results with SOTA baselines and a wide range of datasets.
- Good baseline selection in performance evaluation (used FlashAttention)

**Questions For Authors:**

It would be great if you could give some details about how the token selection works with the state-of-the-art attention frameworks like FlashAttention. Does it require modifying the kernel?

**Relation To Broader Scientific Literature:**

This paper provides a new way of compressing and storing the KV cache. The SVD technique will not only be useful to save the memory usage of the KV cache but also useful in reducing the cost for the KV cache storage use cases.

**Theoretical Claims:**

The theoretical analysis and algorithm design (Section 3.1 and Section 4) make sense.

---

> ### Author Rebuttal · Authors · 2025-03-27
>
> Thank you for the supportive comments and for recognizing the novelty of our method and the thorough evaluations. We hope our detailed clarifications below address the remaining concerns.
>
> ---
>
> ### **Q1: Missing clarification on which dataset was used for observations**
>
> We appreciate the reviewer's careful reading. The observations regarding the low-rank structure of pre-RoPE keys, the high similarity of post-RoPE keys, and the KV cache hit rate plots (Figures 2a, 2b, 5b, and 5c) are based on sequences drawn from PG-19 [1], a real-world long-context dataset. We chose PG-19 to ensure that our structural analyses reflect natural language characteristics rather than synthetic artifacts.
>
> For Figure 5a, the results are based on sequences sampled from RULER-NIAH, consistent with our downstream evaluation tasks.
>
> ---
>
> ### **Q2: How does the token selection interact with state-of-the-art attention frameworks like FlashAttention? Does it require kernel modifications?**
>
> Thank you for the insightful question. In our implementation, token selection is applied prior to the attention computation. At each decoding step, we identify the most relevant KV tokens based on chunk-level attention scores. For efficiency, the chunk-level attention score computation (GEMM + Softmax) is fused into a single kernel. The selected KV entries are then gathered into a compact buffer, which is passed directly to the FlashAttention interface.
>
> Since we only modify the input tensors to attention and not the attention logic itself, no modifications to the FlashAttention kernel are required. This makes ShadowKV easy to integrate into modern inference stacks that already support FlashAttention.
>
> [1] Compressive Transformers for Long-Range Sequence Modelling

---

### Official Review · Reviewer_QmhP · 2025-03-11

**Overall Recommendation:** 3

**Summary:**

This paper presents several interesting findings, including the observation that pre-ROPE keys exhibit low-rank properties, and that post-ROPE keys show high similarity with neighboring tokens.
Based on these insights, the authors propose two main techniques: Low-Rank Keys and Offloaded Values for Storage, and Accurate KV Selection for Fast Decoding.
The designs are well-reasoned and supported by comprehensive experiments.
The proposed methods demonstrate strong compression performance, achieving impressive results across multiple models and long-text benchmarks.

## update after rebuttal
The rebuttal partially addresses my concerns, so I will maintain my rating of 3.

**Claims And Evidence:**

The paper mentions that the reconstruction of the low-rank keys and the movement of values from the CPU to the GPU are synchronized.
As a result, the authors omit this part of the process in their calculation of the Theoretical Equivalent Bandwidth.
However, it would be important to experimentally consider the time overhead of this step.
Specifically, the reconstruction involves an additional matrix multiplication, followed by ROPE which has an O(L) complexity, this might have a significant impact on performance that should be accounted for.

**Essential References Not Discussed:**

Previous works like Eigen-attention[1], HeadsKV[2], MatyroshkaKV[3] has also explored the low-rank properties of the KV cache and utilized corresponding methods, such as SVD/PCA, to reduce the KV cache footprint.

[1] Eigen Attention: Attention in Low-Rank Space for KV Cache Compression
[2] Effectively Compress KV Heads for LLM
[3] MatryoshkaKV: Adaptive KV Compression via Trainable Orthogonal Projection

**Experimental Designs Or Analyses:**

I reviewed the experimental evidence supporting the two key findings, and I find them reasonable.
The selection of RULER and Longbench, which are both long-text datasets, is appropriate for the experimental requirements. However, I believe that the theoretical claims discussed earlier would benefit from some experimental validation to strengthen the argument.

**Methods And Evaluation Criteria:**

The authors propose two main techniques: Low-Rank Keys and Offloaded Values for Storage, which effectively reduce the KV cache footprint for long-text sequences, and Accurate KV Selection for Fast Decoding, which improves accuracy.
The proposed methods are well-designed and supported by comprehensive experiments.
The authors use relevant benchmarks, such as RULER and Longbench, which are highly suitable for the problem at hand, effectively demonstrating the superiority of their approach across multiple models and long-text datasets.

**Other Comments Or Suggestions:**

The overall writing of the paper is well-structured and adheres to proper conventions, with no immediate issues identified.

**Other Strengths And Weaknesses:**

**Strengths**
The paper presents several interesting findings, such as the pre-ROPE key states have a low-rank property compared to other parameters, and that most of the key states post-ROPE exhibit high similarity with neighboring tokens.
These contributions offer valuable perspectives in the field.
Additionally, the proposed methodology is well thought out and highly reasonable.
The detailed results demonstrate strong performance, confirming the effectiveness of the approach.


**Weaknesses**
- The amount of work done in this paper is impressive, particularly the discovery mentioned above. However, the core innovation of the work is not entirely clear. To my knowledge, both the use of SVD decomposition on the KV cache to store low-rank matrices and the application of landmarks for estimating attention scores for selection have been explored in prior work.  What novel adaptations or insights does this paper offer beyond simply combining these existing techniques?

**Questions For Authors:**

I find your findings very intriguing!
However, I have one curiosity regarding the outlier tokens.
Do these outlier tokens correspond to any specific patterns in the original text?
Is there a way to identify these tokens as outliers based on the original text itself?
Additionally, I noticed that you only presented results from a subset of the layers.
Have you analyzed whether the number of outlier tokens correlates with the number of layers and heads in the transformer model?

**Relation To Broader Scientific Literature:**

This paper’s findings, such as pre-Rope keys having the lowest rank and post-Rope keys showing high similarity between adjacent tokens, offer valuable insights for KV cache compression. These results extend prior work on transformer optimizations by highlighting key structural patterns, which can lead to more efficient compression techniques and improved memory and computational performance in large models.

**Theoretical Claims:**

The theoretical proofs provided in Section 4.2 are generally sound.
However, I have some concerns about one of the underlying assumptions: that the time required for low-rank key states reconstruction can be disregarded.
As I mentioned in the "Claims and Evidence" section, I find this assumption somewhat questionable.
Given that the reconstruction involves an additional matrix multiplication, and ROPE has an O(L) complexity, the potential time cost should be considered more thoroughly.

The provided theoretical proofs in Appendix A.2 were reviewed and appear correct and carefully derived, and no immediate issues were identified.

---

> ### Author Rebuttal · Authors · 2025-03-27
>
> Thank you for the thoughtful and thorough review. We truly appreciate your recognition of our findings and experiments. We have thoroughly addressed each of your questions and hope our responses will lead you to consider raising your score.
>
> ---
>
> ### **Q1: Concern about unaccounted latency from key reconstruction and value fetching in the Theoretical Equivalent Bandwidth analysis.**
>
> As detailed in Appendix A.6, we provide a full component-wise decoding latency breakdown (in milliseconds) across various batch sizes and context lengths on an A100 GPU, which includes the time cost of both key reconstruction (Recompute K) and value cache fetching (Fetch V). Our system is designed to leverage CUDA multi-streams to overlap these two operations.
>
> |Context|GEMM+Softmax|Max|TopK|Recompute K (Overlapped)|Fetch V|Attention|FFN|QKV|
> |-|-|-|-|-|-|-|-|-|
> |48$\times$64K|0.56|0.07|0.14|1.25|1.84|0.23|0.33|0.05|
> |24$\times$128K|0.58|0.07|0.15|1.36|1.66|0.21|0.29|0.05|
> |12$\times$256K|0.65|0.07|0.16|1.49|1.75|0.19|0.25|0.05|
> |6$\times$512K|0.71|0.07|0.17|1.51|1.69|0.18|0.23|0.05|
>
> As shown in the table, Fetch V and Recompute K exhibit comparable latency. Since they are executed on separate CUDA streams, their durations can be effectively overlapped—meaning only Fetch V contributes to the critical path.
>
> Furthermore, our measured end-to-end throughput gains validate that these design choices translate into practical system-level improvements.
>
> ---
>
> ### **Q2: Some references not discussed.**
>
> As stated in Section 3.1 (Line 188), prior low-rank approaches primarily focus on **data-independent, offline weight decomposition**, i.e., performing low-rank factorization on the **model weight matrices** using calibration data or during training. These methods either **require training or achieve limited compression**. The cited works (EigenAttention, HeadsKV, and MatryoshkaKV) belong to this category. Specifically:
>
> - EigenAttention generates lower-rank weight matrices, yielding at most 40% KV cache reduction.
> - HeadsKV converts MHA into GQA via low-rank decomposition of weights, which requires training and is only applicable to MHA-based models.
> - MatryoshkaKV trains orthogonal projection matrices, achieving around 60% average KV cache compression.
>
> In contrast, ShadowKV **identifies the dynamic, sequence-dependent low-rank structure in pre-RoPE keys**, and performs **online SVD directly on the KV cache** during inference. This approach is training-free, adaptive to each sequence, and achieves significantly higher compression (over 6$\times$, 15.6% of the original memory).
>
> Thank you for giving us the chance to elaborate on this point. We will add these mentioned references to the future version.
>
> ---
>
> ### **Q3: Core innovation of the work.**
>
> As discussed in Q2, unlike existing low-rank methods that compress weights offline, ShadowKV performs online activation-level SVD on the KV cache in a sequence-adaptive manner, and achieves substantially better compression while preserving accuracy. Prior methods rely on static weight decomposition and often require fine-tuning or model modification.
>
> Moreover, our approach is tightly integrated with a system-level design. We carefully co-design algorithm and system by enabling low-rank key cache reconstruction and CPU-to-GPU value fetching to overlap through CUDA multi-streams. Additionally, we utilize an accurate sparse KV selection mechanism that reduces attention computation and memory movement. This holistic integration ensures end-to-end throughput gains.
>
> ---
>
> ### **Q4: Outlier tokens — are there interpretable patterns? And does outlier count correlate with layers/heads?**
>
> Thank you for the insightful question. We clarify that we selected a subset of layers rather than showing all layers in Figure 5c for visualization purposes to avoid clutter. We performed additional analysis to better understand the nature of outlier tokens and their distribution.
>
> We found that certain tokens, such as attention sinks [1], are consistently identified as outliers, with their similarity scores even being negative, highlighting their distinctiveness in the distribution. Across heads, the sets of outliers tend to differ. We did not observe a clear trend or correlation between outlier frequency and specific layers or head indices.
>
> We absolutely agree that this is a rich direction for future exploration, and we plan to conduct deeper investigations into the interpretability and structure of outliers in future work.
>
> [1] Efficient Streaming Language Models with Attention Sinks

---

### Official Review · Reviewer_vsSH · 2025-03-13

**Overall Recommendation:** 3

**Summary:**

This paper finds a novel method of KV cache management that takes advantage of partial offloading to CPU and matrix decomposition to obtain impressive KV reduction without affecting accuracy and reducing latency significantly. The paper finds two important properties of LLMs that it takes advantage of: (i) the Key cache in KV caches, before applying positional embeddings (this rotates each token/vector in K by an amount determined by its position so that the model can learn positions of tokens), has very low rank and thus high compressibility. (ii) A majority of K tokens after positional embeddings have been applied, show high cosine similarity to adjacent tokens, with few outliers.
What this paper does it takes advantage of property (i) by offloading the V (Value) cache to CPU memory and compress K (key) using Singular Value Decompsition (SVD) cache in GPU memory, thus during inference when KV values are needed, while V is being fetched from CPU simultaneously K is being uncompressed and has positional embeddings applied to it. Furthermore, it takes advantage of (ii) to perform a ‘chunk level approximation’, where a small number of K tokens (1.56%) are retained that can be used to approximate most of the other tokens while the outliers are retained in GPU memory. This leads to further compression. The results show 3x improvement in throughput while supporting 6x larger inputs or batches, this performs better with regards to throughput than a theoretical infinite KV cache memory on GPU, due to minimizing total data movement.

**Claims And Evidence:**

The paper makes modest claims and back them up through empirical evidence.

**Essential References Not Discussed:**

NA

**Experimental Designs Or Analyses:**

The experiments are designed and executed well and the results are analyzed in sufficient depth

**Methods And Evaluation Criteria:**

The evaluation criteria are explained reasonably well.

**Other Comments Or Suggestions:**

This paper proposes a method of KV cache compression and latency improvements obtaining very impressive results with two complementary approaches. It introduces a method of obtaining a very high compression ratio on K caches, and offloading V to CPU. During inference fetching V from CPU memory and K reconstruction are overlapped.
Some things can be addressed to clarify some parts. In figure 8 it looks like the x axis shows KV compression as it is labelled as ‘Sparse KV Cache Budget’ and shows up to 0.20% KV budget while retaining accuracy. This implies up to 500x compression, however the introduction states there is a 6x compression. It seems that ‘Sparse KV Cache Budget’ is referring to something other than total KV cache memory usage and should be clarified.
The authors test the method on 3 models, but this raises the question whether this method can generalize to other models and positional embeddings other than RoPE. This should be discussed in the paper.
Another possible limitation that should be addressed is how the CPU side affects the performance. The authors implicitly assume that fetching V from CPU memory and reconstructing K in GPU memory have around the same time, but what is the CPUs are slow, or CPU to GPU bandwidth is low. This should also be addressed.
A final possible limitation of this paper is how GPU memory usage changes during inference. Outside of inference the authors report 6x compression of GPU memory, but if the KV cache is reconstructed during inference, then it seems that the GPU memory will spike back up to roughly baseline KV cache size. This needs to be addressed in the paper.

**Other Strengths And Weaknesses:**

Strengths:
•	Paper discovers two separate important properties of LLMs and exploits them each, to obtain very impressive compression (6x reduction in GPU memory) and throughput increases (3x higher).
•	Benchmarks are comprehensive, with 3 different benchmarks used, along with 3 different LLM models. Behaviors of different parameters are also shown in the paper. Comparisons to several other state of the art KV methods are also done.

Weaknesses:
•	In Figure 8, x axis needs clarification. The paper reports 6x larger batch sizes, yet the figures here show x axis to have ‘sparse budgets’ down to 0.2% while maintaining performance, which us much more than a 6x compression. It looks like sparse budgets here is referring to something other than KV cache GPU memory usage, this should be clarified in the caption.
•	Some undefined terminology: KV cache hit rate is not defined in text or in the figures; the ‘D’ in D(H1, H2) in observation section is not defined, along with a few other mathematical notations in that section.
•	Authors do not mention how GPU memory usage increases during inference, as the K cache is reconstructed and V cache is prefetched. It seems like the memory usage should spike up during inference in this case.
•	It would be good if the paper briefly addresses the limitations of this approach. Currently it does not.

**Questions For Authors:**

Can you please address the concerns in the weaknesses part above?

**Relation To Broader Scientific Literature:**

The paper does a good job in comparing the proposed idea against existing works.

**Theoretical Claims:**

The paper does not have a theoretical claim.

---

> ### Author Rebuttal · Authors · 2025-03-27
>
> Thank you for detailed review and valuable feedback. We appreciate the reviewer's recognition of the novelty and effectiveness of our method. Below, we address the raised concerns below and will incorporate clarifications into the revised version. We hope the reviewer can consider raising your score in light of our response.
>
> ---
>
> ### **Q1: Clarification on Figure 8 and the meaning of "Sparse KV Cache Budget".**
>
> The x-axis in Figure 8 refers to the percentage of KV pairs selected at each decoding step for attention computation—i.e., the active sparse KV budget per step, rather than the overall GPU memory usage for KV cache storage. In contrast, the 6$\times$ memory compression refers to the total reduction in GPU memory footprint for the KV cache, achieved through a combination of low-rank key compression and value offloading. The extremely small sparse budget (e.g., 1.5%) is made possible by our accurate TopK selection mechanism. We will clarify this distinction explicitly in the figure caption in the future version.
>
> ---
>
> ### **Q2: Undefined terminology such as KV cache hit rate and $D(H_1, H_2)$.**
>
> Thank you for the valuable suggestion. We will add clarifications below in the future version.
>
> - The KV cache hit rate refers to the proportion of selected KV chunks at decoding step $t$ that are also selected at step $t+1$, due to temporal locality in generation. This metric reflects the effectiveness of caching selected sparse KV pairs across decoding steps to reduce redundant key cache reconstruction and value cache fetching.
> - As for $D(H_1, H_2)$, it is defined in Section 3.1 (Line 204) as $D(H_1, H_2) = \langle H_1, H_2 \rangle / r$, where $H_1$ and $H_2$ are rank-$r$ projection matrices and $\langle \cdot, \cdot \rangle$ denotes the Frobenius inner product. We agree the definition could be made more prominent, and we will revise the presentation to improve clarity and visibility.
>
> ---
>
> ### **Q3: GPU memory usage during inference — will it spike due to K reconstruction and V fetching?**
>
> In ShadowKV, GPU memory usage remains bounded and does not spike during inference. This is due to two key factors:
> - At each decoding step, only a sparse subset of KV pairs (e.g., 1.5%) is selected for reconstruction and fetching, not the full sequence.
> - These operations are performed layer-by-layer, and the buffers are reused across layers, so peak memory does not accumulate across the entire model.
>
> For example, in Table 3 (e.g., 122K context, batch size of 24), we reconstruct 1.56% of the full KV cache during each decoding step, resulting in <200MB peak additional memory usage—negligible compared to the full KV size. We will add a memory profiling figure to better illustrate this in the future version.
>
> ---
>
> ### **Q4: The paper currently does not mention limitations.**
>
> Thank you for this suggestion. We plan to add a limitations paragraph, covering the following points:
>
> 1. **Dependency on RoPE positional encoding.** Our low-rank key compression relies on the pre-RoPE key representation being compressible. While RoPE is used in most state-of-the-art LLMs, the method may require adaptation for models using absolute or learned positional embeddings. We are exploring this generalization as future work.
>
> 2. **Linear-time sparse KV selection.** Our current KV selection requires an $O(N)$ complexity. We view the design of $O(\log N)$-time sparse search methods as a promising direction.
>
> 3. **SVD-based compression on low-precision hardware.** ShadowKV relies on SVD for key cache compression, which is highly efficient on modern GPUs. However, on edge devices with only low-precision compute units (e.g., INT8-only cores), efficient SVD implementation may be challenging. Exploring lightweight alternatives to SVD could improve portability to constrained hardware.
>
> ---
>
> ### **Q5: Generalization to non-RoPE models.**
>
> We focus on RoPE-based models because RoPE is the dominant positional encoding scheme in state-of-the-art LLMs (e.g., LLaMA, Qwen, Yi, GLM, etc.). We acknowledge that generalization to other positional embeddings is an interesting direction, and we are currently exploring variants such as ALiBi in ongoing work.
>
> ---
>
> ### **Q6: What if CPU-to-GPU bandwidth is low, or the CPU is slow?**
>
> ShadowKV performs well under typical bandwidth conditions and is designed to keep transfer sizes small and overlappable. We clarify that:
>
> - Value cache fetching is PCIe bandwidth-bound. The CPU merely serves as a memory pool for the value cache and is not involved in computation. Hence, CPU speed has negligible impact on performance.
> - In our implementation, we use PCIe Gen4 with 32 GB/s bandwidth, which is standard in modern server-grade inference deployments. In practice, the PCIe bandwidth is typically well-matched to the GPU performance. For instance, H200 GPUs are commonly paired with PCIe Gen5 (128 GB/s). Therefore, it is rare to find a high-performance GPU configured with a low-performance PCIe.

---

### Official Review · Reviewer_Pdg2 · 2025-03-14

**Overall Recommendation:** 4

**Summary:**

Large language models (LLMs) excel at handling extended contexts but face challenges with key-value (KV) cache scaling, increasing memory usage and reducing inference throughput. Existing methods like KV eviction and sparse attention either degrade accuracy or inadequately optimize memory use and latency. This paper introduces SHADOWKV, leveraging two key insights: (1) pre-RoPE keys are inherently low-rank and highly compressible, and (2) value caches, being less compressible, can be efficiently offloaded to CPU storage. SHADOWKV integrates low-rank key compression, value cache offloading, and precise sparse KV selection to significantly reduce memory usage while preserving high accuracy and low latency. Comprehensive experiments demonstrate SHADOWKV achieves 6× larger batch sizes and throughput gains up to 3× compared to baseline methods across various LLMs and tasks.

**Claims And Evidence:**

SHADOWKV claims significant improvements in accuracy and system performance through its combination of low-rank compression and optimized memory management. The evidence supporting accuracy improvements is robust, detailed, and quantitatively convincing. The paper also provides adequate quantitative analysis regarding the potential benefits to system performance, though further detailed experiments on throughput could strengthen these claims.

**Essential References Not Discussed:**

None explicitly identified.

**Experimental Designs Or Analyses:**

The experimental design, primarily focused on accuracy and system performance metrics, is solid. Accuracy evaluations are especially thorough and convincing. However, throughput experiments could benefit from more comprehensive scenarios or additional benchmarks to better understand the practical limits and generalizability of the proposed approach.

**Methods And Evaluation Criteria:**

The paper evaluates two primary metrics: accuracy and throughput. Accuracy evaluation is extensive, clearly demonstrating the effectiveness of the proposed compression techniques. Throughput evaluations, while less thorough, sufficiently illustrate efficiency improvements. Expanding throughput analysis, particularly across more diverse settings, would further strengthen the evaluation.

**Other Comments Or Suggestions:**

Overall, this is a well-written and carefully prepared paper.

**Other Strengths And Weaknesses:**

A notable strength of the paper is its insightful design, particularly in efficiently compressing pre-RoPE key caches and significantly reducing CPU-GPU communication overhead, demonstrating substantial potential for long-context scenarios.

However, the evaluation of throughput is somewhat limited. Memory savings do not always translate directly into performance gains; thus, expanding this evaluation with additional context or detailed performance metrics could enhance the credibility of system-related claims.

**Questions For Authors:**

1. In a typical scenario, what is the time breakdown for low-rank key cache reconstruction?

2. How effectively can value cache fetching overlap with computation, and under what conditions?

3. Could you clarify the experimental settings used for generating Table 3? Specifically, is this based on standard vLLM?

4. What are the primary limitations that might prevent integrating SHADOWKV into modern LLM serving engines like vLLM and SGLang?

**Relation To Broader Scientific Literature:**

The paper situates itself within existing literature on KV cache management strategies, such as compression and offloading techniques, highlighting its contribution with the insight on compressing RoPE keys and reconstruct values.

**Theoretical Claims:**

The theoretical foundation for low-rank key compression is plausible and well-articulated, though I am less confident in fully assessing its theoretical rigor. Nevertheless, quantitative system performance analyses and experimental results effectively support these theoretical claims.

---

> ### Author Rebuttal · Authors · 2025-03-27
>
> Thank you for the thoughtful review and your positive assessment of our work. We appreciate your recognition of the novelty of ShadowKV, particularly its system-level insights and the strength of accuracy evaluations. We address your concerns with additional clarifications and experiments as detailed below.
>
> ---
>
> ### **Q1: Throughput evaluation is somewhat limited. Memory savings do not always translate directly into performance gains.**
>
> We appreciate the suggestion. To further support our claim, we include a table below, showing throughput (tokens/s) under varying batch sizes and contexts for both Full KV and ShadowKV with Llama3-8B-1M.
>
> |(seq,method)|bsz=1|bsz=2|bsz=3|bsz=4|bsz=5|bsz=6|bsz=8|bsz=12|bsz=16|bsz=24|bsz=32|bsz=48|
> |-|-|-|-|-|-|-|-|-|-|-|-|-|
> |60K,Full|58.62|89.19|111.44|126.73|142.62|147.40|**160.62**|OOM|OOM|OOM|OOM|OOM|
> |122K,Full|45.59|65.12|75.16|**80.77**|OOM|OOM|OOM|OOM|OOM|OOM|OOM|OOM|
> |244K,Full|32.22|**40.37**|OOM|OOM|OOM|OOM|OOM|OOM|OOM|OOM|OOM|OOM|
> |488K,Full|**20.15**|OOM|OOM|OOM|OOM|OOM|OOM|OOM|OOM|OOM|OOM|OOM|
> |60K,ShadowKV|47.06|89.69|126.61|159.41|184.92|205.41|244.20|306.09|346.34|399.65|428.63|**455.14**|
> |122K,ShadowKV|36.57|65.61|94.28|115.01|132.23|143.77|166.72|196.73|217.24|**239.51**|OOM|OOM|
> |244K,ShadowKV|27.63|48.39|65.95|78.92|87.83|94.07|104.73|**119.01**|OOM|OOM|OOM|OOM|
> |488K,ShadowKV|17.27|29.82|41.01|47.13|50.85|**53.46**|OOM|OOM|OOM|OOM|OOM|OOM|
>
> - ShadowKV enables significantly larger batch sizes at long context lengths, where Full KV quickly runs out of memory.
> - At small batch sizes (e.g., bsz = 1), ShadowKV may be slightly slower due to reconstruction and fetching overhead. However, as batch size increases, the benefit of memory savings and increased parallelism translate directly into throughput gains.
>
> These additional results help clarify that ShadowKV not only reduces memory usage but also yields consistent throughput improvements. We will include it in the final version and appreciate the reviewer's suggestion, which helped us strengthen this aspect of the paper.
>
> ---
>
> ### **Q2: What is the time breakdown for low-rank key cache reconstruction in a typical scenario?**
>
> As detailed in Appendix A.6, we provide a full component-wise decoding latency breakdown (in milliseconds) across various batch sizes and context lengths on an A100 GPU, which includes the time cost of both key reconstruction (Recompute K) and value cache fetching (Fetch V). Our system is designed to leverage CUDA multi-streams to overlap these two operations.
>
> |Context|GEMM+Softmax|Max|TopK|Recompute K (Overlapped)|Fetch V|Attention|FFN|QKV|
> |-|-|-|-|-|-|-|-|-|
> |48$\times$64K|0.56|0.07|0.14|1.25|1.84|0.23|0.33|0.05|
> |24$\times$128K|0.58|0.07|0.15|1.36|1.66|0.21|0.29|0.05|
> |12$\times$256K|0.65|0.07|0.16|1.49|1.75|0.19|0.25|0.05|
> |6$\times$512K|0.71|0.07|0.17|1.51|1.69|0.18|0.23|0.05|
>
> ---
>
> ### **Q3: How effectively can value cache fetching overlap with computation, and under what conditions?**
>
> As discussed in Appendix A.6, we leverage CUDA multi-streams to overlap CPU-GPU data transfers (Fetch V) with GPU-side compute (Recompute K), ensuring high GPU utilization during decoding. This overlap is particularly effective in long-context settings where attention computation dominates latency.
>
> As shown in the table, Fetch V and Recompute K exhibit comparable latency. Since they are executed on separate CUDA streams, their durations can be effectively overlapped—meaning only Fetch V contributes to the critical path.
>
> Furthermore, our measured end-to-end throughput gains validate that these design choices translate into practical system-level improvements.
>
> ---
>
> ### **Q4: Experimental settings for Table 3—are they based on standard vLLM?**
>
> Our baseline implementation leverages open-source efficient CUDA kernels, including those from both vLLM and FlashInfer [1]. Our inference engine is optimized specifically for large-batch long-context decoding.
>
> The table below presents a decoding latency (tokens/s) benchmark, demonstrating that our baseline achieves comparable or slightly better efficiency than vLLM in full-attention mode. This shows that our baseline is a valid and fair reference point.
>
> |Context (Full Attention)|vLLM |Our Baseline|
> |-|-|-|
> |1$\times$30K|64.19|69.38|
> |1$\times$60K|55.30|58.62|
> |1$\times$120K|43.41|46.60|
> |1$\times$240K|30.51|32.78|
> |1$\times$480K|19.04|20.74|
>
> ---
>
> ### **Q5: Limitations for integrating SHADOWKV into vLLM and SGLang?**
>
> ShadowKV employs custom CUDA kernels for low-rank reconstruction, RoPE fusion, and efficient CPU data prefetching with overlap , built on top of CUTLASS [2]. Integrating ShadowKV into vLLM and SGLang would require backend modifications to support these kernels and its memory pool management strategy. While this integration is non-trivial, it is technically feasible, and we are actively working toward supporting popular serving engines in the future.
>
> [1] https://github.com/flashinfer-ai/flashinfer
>
> [2] https://github.com/NVIDIA/cutlass

---

> > ### Comment · Reviewer_Pdg2 · 2025-04-02
> >
> > Thank you for the detailed explanation I do not have any further questions, thank you for the good work again : )

---

> > > ### Author Response · Authors · 2025-04-03
> > >
> > > Thank you for your kind comments and encouraging feedback. We are pleased to hear that our responses were helpful and truly appreciate your support of our work.

---

### Decision · Program_Chairs · 2025-05-01

**Decision:**

Accept (spotlight poster)

**Comment:**

This paper aims to reduce the memory overhead of long-context LLM inference. It discovers a low-rank property of key vectors before the positional encodings are applied, which leads to a novel offloading algorithm. Evaluations across various models and tasks are comprehensive and well-executed.

All reviewers agree that the paper novel, well-motivated, and well-exacted. Most concerns have been addressed.